# Weight-Balancing Fixes and Flows for Deep Learning

**Lawrence K. Saul**                                                          *lsaul@flatironinstitute.org*
*Flatiron Institute, Center for Computational Mathematics*
*162 Fifth Avenue, New York, NY 10010*

**Reviewed on OpenReview:** *https://openreview.net/forum?id=uaHyXxyp2r*

## Abstract

Feedforward neural networks with homogeneous activation functions possess an internal symmetry: the functions they compute do not change when the incoming and outgoing weights at any hidden unit are rescaled by reciprocal positive values. This paper makes two contributions to our understanding of these networks. The first is to describe a simple procedure, or *fix*, for balancing the weights in these networks: this procedure computes multiplicative rescaling factors—one at each hidden unit—that rebalance the weights of these networks without changing the end-to-end functions that they compute. Specifically, given an initial network with arbitrary weights, the procedure determines the functionally equivalent network whose weight matrix is of minimal $\ell_{p,q}$-norm; the weights at each hidden unit are said to be balanced when this norm is stationary with respect to rescaling transformations. The optimal rescaling factors are computed in an iterative fashion via simple multiplicative updates, and the updates are notable in that (a) they do not require the tuning of learning rates, (b) they operate in parallel on the rescaling factors at all hidden units, and (c) they converge monotonically to a global minimizer of the $\ell_{p,q}$-norm. The paper's second contribution is to analyze the optimization landscape for learning in these networks. We suppose that the network's loss function consists of two terms—one that is invariant to rescaling transformations, measuring predictive accuracy, and another (a regularizer) that breaks this invariance, penalizing large weights. We show how to derive a weight-balancing *flow* such that the regularizer remains minimal with respect to rescaling transformations as the weights descend in the loss function. These dynamics reduce to an ordinary gradient flow for $\ell_2$-norm regularization, but not otherwise. In this way our analysis suggests a canonical pairing of alternative flows and regularizers.

## 1 Introduction

Many recent studies of deep learning have focused on the important role of symmetries (Bronstein et al., 2021; Kunin et al., 2021; Tanaka & Kunin, 2021; Gluch & Urbanke, 2021; Armenta & Jodoin, 2021). In large part these studies were inspired by the role that symmetries play in our understanding of the physical world (Anderson, 1972; Zee, 2016). Of particular interest are symmetries that arise when a model is formulated or expressed in terms of more parameters than its essential degrees of freedom. In physics these symmetries arise from so-called *gauge* degrees of freedom (Gross, 1992), while in deep learning they are present in many popular models of overparameterized networks.

One such model in machine learning is a feedforward network with rectified linear hidden units. Such a network is specified by the values of its weights, but the function it computes does not change when the incoming and outgoing weights at any hidden unit are inversely rescaled by some positive value (Glorot et al., 2011); see Fig. 1. This particular symmetry of deep learning has already led to many important findings. For example, it is known that this symmetry gives rise to a conservation law: at each hidden unit, there is a certain balance of its incoming and outgoing weights that does not change when these networks are trained by gradient flow (i.e., gradient descent in the limit of an infinitesimally small learning rate) (Du et al., 2018). From this conservation law follows another important observation: if the weights are initially

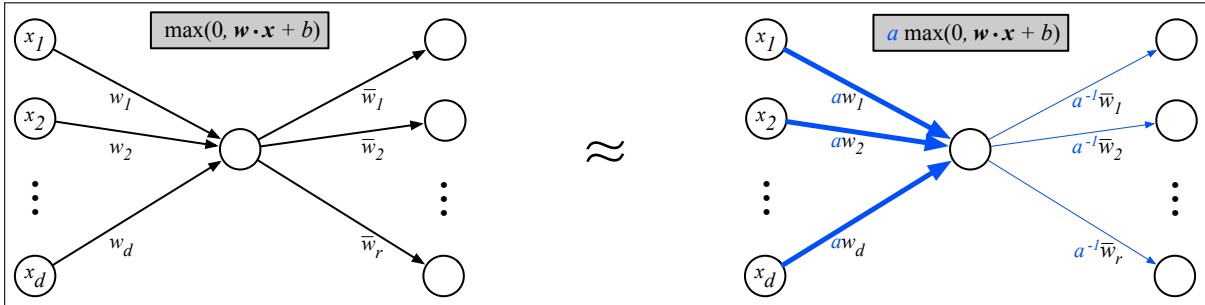

Figure 1: A rectified linear unit (ReLU) has the same effect if its incoming weights $\mathbf{w}$ and bias $b$ are rescaled by some factor $a > 0$ while its outgoing weights $\overline{\mathbf{w}}$ are rescaled by the inverse factor $a^{-1}$. When $a > 1$, the rescaling increases the magnitudes of incoming weights and decreases the magnitudes of outgoing weights; when $a < 1$, the effects are reversed.

balanced across layers, then they remain so during training, a key condition for proving certain convergence results (Arora et al., 2019). It is also possible, by analyzing the synaptic flows across adjacent layers, to devise more powerful pruning algorithms (Tanaka et al., 2020). Finally, a number of authors have proposed more sophisticated forms of learning that are invariant to these rescaling transformations (Neyshabur et al., 2015a; Meng et al., 2019) or that break this invariance in purposeful ways (Badrinarayanan et al., 2015; Stock et al., 2019; Armenta et al., 2021; Zhao et al., 2022).

These latter works highlight an important distinction: though rescaling transformations do not change the functions computed by these networks, they may affect other aspects of learning. For example, unbalanced networks may take longer to converge and/or converge to altogether different solutions. It is therefore important to understand the different criteria that can be used to break this invariance.

## 1.1 Inspiration from physics

To proceed, we consider how similar ideas have been developed to study the physical world. The simplest gauge symmetry arises in classical electrodynamics (Jackson, 2002). The gauge degrees of freedom in this setting may be *fixed* to highlight certain physics of interest. Two well-known choices are the Coulomb gauge, which minimizes the volume integral of the squared vector potential (Gubarev et al., 2001), and the Lorenz gauge, which simplifies certain wave equations. It is natural to ask if there are analogous fixes for deep neural networks—fixes, for instance, that minimize the norms of weights or simplify the dynamics of learning.

With these goals in mind, this paper investigates a family of norm-minimizing rescaling transformations for feedforward networks with homogeneous activation functions. These transformations are designed to minimize the entri-wise $\ell_{p,q}$-norm (with $p, q \geq 1$) of the network's weight matrix $\mathbf{W}$, defined as

$$\|\mathbf{W}\|_{p,q} = \left( \sum_i \left( \sum_j |W_{ij}|^p \right)^{\frac{q}{p}} \right)^{\frac{1}{q}}, \tag{1}$$

without changing the end-to-end function that the network computes (Neyshabur et al., 2015a). Here $W_{ij}$ is the incoming weight at unit $i$ from unit $j$. A particular transformation is specified by a set of multiplicative rescaling factors, one at each hidden unit of the network. *The first main contribution of this paper is to show how to obtain rescaling factors that minimize the norm in eq. (1).* More concretely, we derive simple multiplicative updates to compute these factors (Algorithm 1), and we prove that these updates converge monotonically to fixed points that represent minimum-norm solutions (Theorem 1). Notably, these multiplicative updates are parameter-free in that they do not require the setting or tuning of learning rates.

### 1.2 From fixes to flows

Norm-minimizing rescaling transformations provide one way to balance the incoming and outgoing weights at each hidden unit in the network. We also analyze the optimization landscape in networks with rescaling symmetries and consider how the balancedness of weights can be preserved during learning—that is, in the course of adapting the network's weights and biases.

This analysis yields an interesting counterpart to Noether's theorem for physical systems (Noether, 1918). When a physical system is described by a Lagrangian, its dynamics are specified by a least action principle, and Noether's theorem gives a recipe to deduce the conserved quantities that are generated by its symmetries. The theorem has also been applied to deduce conserved quantities in deep learning (Kunin et al., 2021; Tanaka & Kunin, 2021; Gluch & Urbanke, 2021). The networks we study in this paper have a symmetry group of rescaling transformations, but here we have sought to solve a sort of *inverse* problem. Specifically, in our case, the conserved quantities are specified by the choice of regularizer, and instead we have needed a recipe to deduce a compatible dynamics for learning—that is, one in which the desired quantities are conserved.

It is easy to see how this inverse problem arises. Let $\mathcal{E}(\boldsymbol{\Theta})$ measure the training error of a network in terms of its weights and biases, denoted collectively by $\boldsymbol{\Theta} = (\mathbf{W}, \mathbf{b})$. Also, let $\mathcal{R}(\mathbf{W})$ denote a regularizer that penalizes large weights in some way. When this regularizer is minimized with respect to rescaling transformations, it induces a particular balance of the incoming and outgoing weights at each hidden unit of the network; see Fig. 1. This minimality is expressed by the zeros of derivatives with respect to rescaling factors at each hidden unit, and these zeros are the quantities that we seek to conserve for a weight-balancing flow. Thus we start from a regularizer $\mathcal{R}(\mathbf{W})$ and seek a dynamics that not only *conserves* these zeros, but also *descends* in a loss function of the form $\mathcal{L}(\boldsymbol{\Theta}) = \mathcal{E}(\boldsymbol{\Theta}) + \lambda\mathcal{R}(\mathbf{W})$ for any $\lambda \geq 0$.

*The second main contribution of this paper is to give a recipe for such dynamics—namely, to derive weight-balancing flows such that a regularizer $\mathcal{R}(\mathbf{W})$ remains minimal with respect to rescaling transformations throughout the course of learning.* This recipe is given by Theorem 4, and it has an especially simple form

$$\frac{d}{dt}\left(W_{ij}\frac{\partial \mathcal{R}}{\partial W_{ij}}\right) = -W_{ij}\frac{\partial \mathcal{L}}{\partial W_{ij}}. \tag{2}$$

Additionally, in Theorem 5, we prove that these flows descend under fairly general conditions in the loss function, $\mathcal{L}(\boldsymbol{\Theta})$. The dynamics in eq. (2) reduces to an ordinary gradient flow, namely $\dot{\mathbf{W}} = \frac{1}{2}(\partial\mathcal{L}/\partial\mathbf{W})$, when the weights are regularized by their $\ell_2$-norm (i.e., taking $\mathcal{R}(\mathbf{W}) = \frac{1}{2}\sum_{ij} W_{ij}^2$), but it suggests a much richer set of flows for other regularizers, including but not limited to those based on the $\ell_{p,q}$-norm in eq. (1).

The paper is organized as follows. Section 2 presents the multiplicative updates for minimizing the $\ell_{p,q}$-norm in eq. (1) with respect to rescaling transformations. Section 3 derives the weight-balancing flows in eq. (2). Finally, section 4 relates these ideas to previous work, and section 5 discusses directions for future research.

## 2 Weight balancing

In this section we show how to balance the weights in feedforward networks with homogeneous activation functions without changing the end-to-end function that these networks compute. Section 2.1 describes the symmetry group of rescaling transformations in these networks, and section 2.2 presents multiplicative updates to optimize over the elements of this group. Finally, section 2.3 demonstrates empirically that these updates converge quickly in practice. We refer the reader to appendix A for a formal proof of convergence.

### 2.1 Preliminaries

Our interest lies in feedforward networks that parameterize vector-valued functions $f : \Re^d \to \Re^r$. The following notation will be useful. We denote the indices of a network's input, hidden, and output units, respectively, by $\mathcal{I}$, $\mathcal{H}$, and $\mathcal{O}$, and we order the units so that $\mathcal{I} = \{1, \ldots, d\}$, $\mathcal{H} = \{d+1, \ldots, n-r\}$, and $\mathcal{O} = \{n-r+1, \ldots, n\}$ where $n$ is the total number of units. Let $\mathbf{x} \in \Re^d$ denote an input to the network and $f(\mathbf{x}) \in \Re^r$ its corresponding output. The mapping $\mathbf{x} \to f(\mathbf{x})$ is determined by the network's weight matrix $\mathbf{W} \in \Re^{n \times n}$, biases $\mathbf{b} \in \Re^n$, and activation function $g : \Re \to \Re$ at each non-input unit. The mapping $\mathbf{x} \to \mathbf{f}(\mathbf{x})$

can be computed by the feedforward procedure that sequentially activates all of the units in the network—that is, setting $h_i = x_i$ for the input units, propagating $h_i = g(\sum_j W_{ij}h_j + b_i)$ to the remaining units, and setting $f_i(\mathbf{x}) = h_{n+i-r}$ from the $i^{\text{th}}$ output unit. Since the network is feedforward, its weight matrix is strictly lower triangular, and many of its lower triangular entries are also equal to zero (e.g., between input units). We also assume that the output units are unconnected (with $W_{ij}=0$ if $j \in \mathcal{O}$ and $i > j$) and that the same activation function is used at every hidden unit.

A rescaling symmetry arises at each hidden unit when its activation function is positive homogeneous of degree one (Neyshabur et al., 2015a; Dinh et al., 2017)—that is, when it satisfies $g(az) = ag(z)$ for all $a > 0$; see Fig. 1. In this paper we focus on networks of rectified linear hidden units (ReLUs), with the activation function $g(z) = \max(0, z)$ at all $i \in \mathcal{H}$, but this property is also satisfied by linear units (Saxe et al., 2013), leaky ReLUs (He et al., 2015), and maxout units (Goodfellow et al., 2013).

Our results in this paper apply generally to feedforward networks (i.e., those represented by directed graphs) with homogeneous activation functions. For obvious reasons, the case of layered architectures is of special interest, and in such architectures, the weight matrix will have many zeros indicating the absence of weights between non-adjacent layers. However, we will not assume this structure in what follows. Our only assumption is that in each network there are no *vacuous* hidden units—or equivalently, that every hidden unit of the network plays some role in the computation of mapping $\mathbf{x} \to \mathbf{f}(\mathbf{x})$. For example, a minimal requirement is that for each $i \in \mathcal{H}$, there exists some $j < i$ such that $W_{ij} \neq 0$ and also some $j > i$ such that $W_{ji} \neq 0$.

Next we consider the symmetry group of rescaling transformations in these networks. Each such transformation is specified by a set of rescaling factors, one for each hidden unit. We use

$$\mathcal{A} = \{\mathbf{a} \in \Re^n \,|\, a_i > 0 \text{ if } i \in \mathcal{H}, a_i = 1 \text{ otherwise}\} \tag{3}$$

to denote the set of these transformations. Then under a particular rescaling, represented by some $\mathbf{a} \in \mathcal{A}$, the network's weights and biases are transformed multiplicatively as

$$W_{ij} \leftarrow W_{ij}(a_i/a_j), \tag{4}$$
$$b_i \leftarrow b_i a_i. \tag{5}$$

It may seem redundant in eq. (3) to introduce rescaling factors at non-hidden units only to constrain them to be equal to one. With this notation, however, we can express the transformations in eqs. (4–5) without distinguishing between the network's different types of units. As shorthand, we write $(\mathbf{W'}, \mathbf{b'}) \sim (\mathbf{W}, \mathbf{b})$ and $\mathbf{W'} \sim \mathbf{W}$ to denote when these parameters are equivalent via eqs. (4–5) up to some rescaling.

## 2.2 Multiplicative updates

The first goal of this paper is to solve the following problem. Given a feedforward network with weights $\mathbf{W}_0$ and biases $\mathbf{b}_0$, we seek the functionally equivalent network, with $(\mathbf{W}, \mathbf{b}) \sim (\mathbf{W}_0, \mathbf{b}_0)$, such that $\mathbf{W}$ has the smallest possible norm in eq. (1). We provide an iterative solution to this problem in Algorithm 1, which computes $\mathbf{W}$ and $\mathbf{b}$ via a *sequence* of rescaling transformations. Each rescaling transformation is in turn implemented by a multiplicative update of the form in eqs. (4–5). For each update, the key step is to compute the rescaling factor $a_i$ at each hidden unit $i \in \mathcal{H}$ from a ratio comparing the magnitudes of its ingoing and outgoing weights. In this section, we describe the basic ingredients of this algorithm.

To begin, we define two intermediate quantities that play important roles in both the multiplicative updates of this section and the weight-balancing flows of section 3.3. The first of these is the *per-unit regularizer*

$$\rho_i(\mathbf{W}) = \left(\sum_j |W_{ij}|^p\right)^{\frac{q}{p}}, \tag{6}$$

which we use to denote the local contribution to the norm in eq. (1) from the incoming weights at the $i^{\text{th}}$ unit in the network. The second of these quantities is the *stochastic matrix*

$$\pi_{ij}(\mathbf{W}) = \frac{|W_{ij}|^p}{\sum_k |W_{ik}|^p} \tag{7}$$

whose columns sum to one. Note that these quantities depend on the values of $p$ and $q$ in the $\ell_{p,q}$-norm, but to avoid an excess of notation, we will not explicitly indicate this dependence.

We now explain the roles played by these quantities in Algorithm 1. In this algorithm, the key step per update is to compute the rescaling factor $a_i$ at each hidden unit of the network. In terms of the above quantities, this rescaling factor takes the especially simple form

$$a_i = \left[ \frac{\sum_j \rho_j(\mathbf{W}) \, \pi_{ji}(\mathbf{W})}{\rho_i(\mathbf{W})} \right]^{\frac{1}{4 \max(p,q)}}. \tag{8}$$

Intuitively, eq. (8) shows that the multiplicative updates have a fixed point (where $a_i = 1$ for all $i$) when the per-unit-regularizer in eq. (6) is a stationary distribution of the stochastic matrix in eq. (7). Finally, we note that many of the computations in eqs. (6–8) can be easily parallelized, and also that the updates in eqs. (4–5) can be applied in parallel after computing the rescaling factors in eq. (8).

We derive the form of these rescaling factors (and particularly, the curious value of their outer exponent) in appendix A. Our first main result is contained in the following theorem:

**Theorem 1** (Convergence of Multiplicative Updates). *For all $\mathbf{W}_0 \in \Re^{n \times n}$ and $\mathbf{b}_0 \in \Re^n$, the multiplicative updates in Algorithm 1 converge to a global minimizer of the entry-wise $\ell_{p,q}$-norm*

$$\underset{\mathbf{W}, \mathbf{b}}{\operatorname{argmin}} \|\mathbf{W}\|_{p,q} \quad \text{such that} \quad (\mathbf{W}, \mathbf{b}) \sim (\mathbf{W}_0, \mathbf{b}_0) \tag{9}$$

*without changing the function computed by the network. Also, the intermediate solutions from these updates yield monotonically decreasing values for these norms.*

A proof of this theorem can also be found in appendix A. Note that Theorem 1 provides stronger guarantees than exist for the full problem of learning via back-propagated gradients. In particular, these weight-balancing updates converge monotonically to a global minimizer, and they do not require the tuning of hyperparameters such as learning rates. One might hope, therefore, that such fixes could piggyback on top of existing learning algorithms without much extra cost, and this has indeed served as motivation for many previous studies of symmetry and symmetry-breaking in deep learning (Neyshabur et al., 2015a; Badrinarayanan et al., 2015; Meng et al., 2019; Stock et al., 2019; Armenta et al., 2021; Zhao et al., 2022).

### 2.3   Demonstration of convergence

Theorem 1 states that the multiplicative updates in Algorithm 1 converge *asymptotically* to a global minimum of the $\ell_{p,q}$-norm. But how fast do they converge in practice? Fig. 2 plots the convergence of the multiplicative updates in Algorithm 1 for different values of $p$ and $q$ and for three randomly initialized networks with differing numbers of hidden layers but the same overall numbers of input (200), hidden (3750), and output (10) units. From shallowest to deepest, the networks had 200-2500-1250-10 units, 200-2000-1000-500-250-10 units, and 200-1000-750-750-500-500-250-10 units. The networks were initialized with zero-valued biases and zero-mean Gaussian random weights whose variances were inversely proportional to the fan-in at each unit (He et al., 2015). The panels in the figure plot the ratio $\|\mathbf{W}\|_{p,q} / \|\mathbf{W}_0\|_{p,q}$ as a function of the number of multiplicative updates, where $\|\mathbf{W}_0\|_{p,q}$ and $\|\mathbf{W}\|_{p,q}$ are respectively the $\ell_{p,q}$-norms, defined in eq. (1), of the initial and updated weight matrix. Results are shown for several values for $p$ and $q$.

As expected, the updates take longer to converge in deeper networks (where imbalances must propagate through more layers), but in general a high degree of convergence is obtained for a modest number of iterations. The panels show that conventionally initialized networks are (i) far from minimal as measured by the $\ell_{p,q}$-norm of their weights and (ii) easily rebalanced by a sequence of rescaling transformations. Finally we note that the results in Fig. 2 did not depend sensitively on the value of the random seed.

---

**Algorithm 1** Given a network with weights $\mathbf{W}_0$ and biases $\mathbf{b}_0$, this procedure returns a functionally equivalent network whose rescaled weights $\mathbf{W}$ and biases $\mathbf{b}$ minimize the norm $\|\mathbf{W}\|_{p,q}$ in eq. (1) up to some tolerance $\delta > 0$. The set $\mathcal{H}$ contains the indices of the network's hidden units. The rescaled weights and biases are computed via a sequence of multiplicative updates of the form in eqs. (4–5).

---

**procedure** $(\mathbf{W}, \mathbf{b}) = \text{MINNORMFIX}(\mathbf{W}_0, \mathbf{b}_0, \mathcal{H}, p, q, \delta)$
    $(\mathbf{W}, \mathbf{b}, \mathbf{a}) \leftarrow (\mathbf{W}_0, \mathbf{b}_0, \mathbf{1})$                                      ▷ *Initialize.*
    **repeat**
        **for all** $(i, j)$ **do**
            $\pi_{ij}(\mathbf{W}) \leftarrow \dfrac{|W_{ij}|^p}{\sum_k |W_{ik}|^p}$                     ▷ *Compute stochastic matrix.*
        **for all** $i$ **do**
            $\rho_i \leftarrow \left( \sum_j |W_{ij}|^p \right)^{\frac{q}{p}}.$                 ▷ *Compute per-unit regularizers.*
        **for all** $i \in \mathcal{H}$ **do**
            $a_i \leftarrow \left[ \dfrac{\sum_j \rho_j(\mathbf{W})\, \pi_{ji}(\mathbf{W})}{\rho_i(\mathbf{W})} \right]^{\frac{1}{4\max(p,q)}}$          ▷ *Compute rescaling factors.*
        **for all** $(i, j)$ **do**
            $W_{ij} \leftarrow W_{ij}(a_i/a_j)$                        ▷ *Rescale weights.*
        **for all** $i$ **do**
            $b_i \leftarrow b_i a_i$                             ▷ *Rescale biases.*
            $\delta_i \leftarrow |a_i - 1|$
    **until** $\max_i(\delta_i) < \delta$                                 ▷ *Iterate until convergence.*

---

## 3 Learning and regularization

The rescaling symmetry in Fig. 1 also has important consequences for learning in ReLU networks (Kunin et al., 2021; Gluch & Urbanke, 2021; Armenta & Jodoin, 2021; Neyshabur et al., 2015a; Meng et al., 2019; Badrinarayanan et al., 2015; Armenta et al., 2021; Zhao et al., 2022). The goal of learning is to discover the weights and biases that minimize the network's loss function on a data set of training examples. In this section we examine the conditions for learning under which some norm or regularizer of the weight matrix (e.g., the $\ell_{p,q}$-norm) remains minimal with respect to rescaling transformations. Equivalently, these are the conditions for learning under which the incoming and outgoing weights at each hidden unit remain *balanced* with respect to this norm. Our interest lies mainly in the following question: are there conditions such that the weight-balancing procedure of the last section can be *one and done* at the outset of learning? To answer this question, we must understand whether learning and weight-balancing are complementary procedures or whether they are operating in some way at cross-purposes (e.g., the former undoing the latter).

Our ultimate goal in this section is to derive the *weight-balancing flows* in eq. (2). Section 3.1 reviews the basic properties of gradient flow, while section 3.2 derives the more general flows for weight-balancing. Our main results, stated in Theorems 4 and 5, are that these flows descend in a regularized loss function while preserving the minimality of the regularizer with respect to rescaling transformations. Finally, section 3.3 analyzes the specific forms and properties of these flows for regularizers based on the $\ell_{p,q}$-norm in eq. (1).

### 3.1 Gradient flow

There are many forms of learning in deep networks. Perhaps the simplest to analyze is gradient flow (Elkabetz & Cohen, 2021), in which the network's parameters are adapted in continuous time along the negative gradient of the loss function. Gradient flows can be derived for any differentiable loss function, and they are widely used to study the behavior of gradient descent in the limit of small learning rates.

There are two properties of gradient flow in neural networks that we will need to generalize for the minimality-preserving flows in eq. (2). The first is the property of *descent*—namely, that the loss function of a network decreases over time. This is simple to demonstrate for gradient flow. Let $\boldsymbol{\Theta} = (\mathbf{W}, \mathbf{b})$ denote the weights and biases of the network, and suppose that the network is trained to minimize a loss function $\mathcal{L}(\boldsymbol{\Theta})$. Then

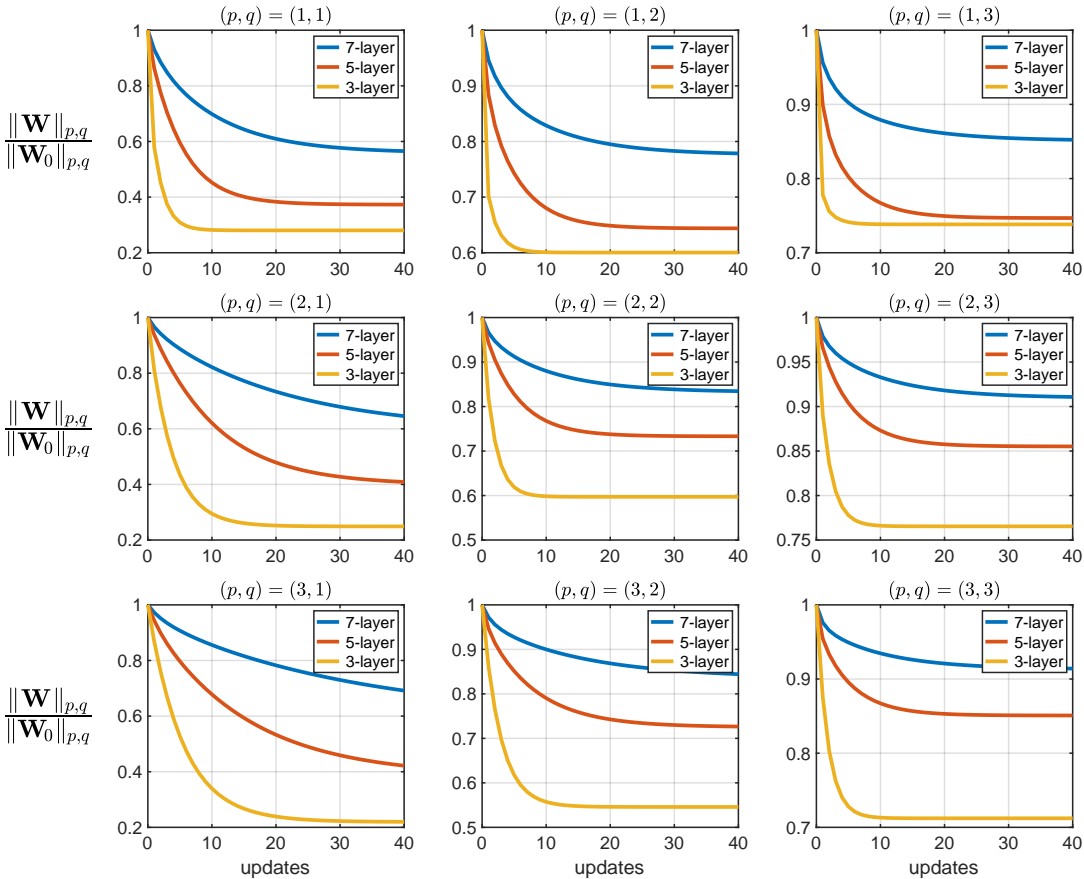

Figure 2: Convergence of weight-balancing multiplicative updates in networks with differing numbers of hidden layers but the same overall numbers of units. The updates minimize the $\ell_{p,q}$-norm of the weight matrix, and each panel shows the results for a different set of values for $p$ and $q$. See text for details.

under gradient flow, at non-stationary points of the loss function, we see that

$$\frac{d\mathcal{L}}{dt} = \frac{\partial \mathcal{L}}{\partial \Theta} \cdot \frac{d\Theta}{dt} = -\frac{\partial \mathcal{L}}{\partial \Theta} \cdot \frac{\partial \mathcal{L}}{\partial \Theta} < 0. \tag{10}$$

From eq. (10) it is also clear that the fixed points of gradient flow correspond to stationary points of the loss function. In the next section, we will see how these properties are preserved in more general flows.

Another important property of gradient flow emerges in feedforward networks with homogeneous activation functions. This property is a consequence of the rescaling symmetry in Fig. 1; when such a network is trained by gradient flow, their rescaling symmetries give rise to *conservation laws*, one at each hidden unit of the network (Du et al., 2018; Kunin et al., 2021; Bronstein et al., 2021; Gluch & Urbanke, 2021; Armenta & Jodoin, 2021). These conservation laws do not depend on the details of the network's loss function, requiring only that it is also invariant[1] under the symmetry group of rescaling transformations. In particular, for any such loss function, it is known that the quantity

$$\Delta_i = b_i^2 + \sum_j W_{ij}^2 - \sum_j W_{ji}^2 \tag{11}$$

is conserved under gradient flow at each hidden unit $i \in \mathcal{H}$. The connection between symmetries and conservation laws is well known in physical systems from Noether's theorem (Noether, 1918), but it is worth noting that the dynamics of gradient flow were not historically derived from a Lagrangian.

---

[1]Typical regularizers break this invariance, but it is possible to construct those that do not (Neyshabur et al., 2015a).

Our next step is to derive the conserved quantities in eq. (11) as a special case of a more general framework. For now, we wish simply to emphasize the logic that has been used to derive many conservation laws in neural networks: to start, one assumes that the network is trained by gradient flow in its weights and biases, and then, inspired by Noether's theorem in physical systems, one derives the conserved quantities that are implied by the network's symmetries. In the next section, we shall flip this script on its head.

## 3.2 Weight-balancing flows

In this section we investigate a larger family of flows for feedforward networks with homogeneous activation functions. We assume that the loss functions for these networks take a standard form with two competing terms—one that measures the network's predictive accuracy on the training data, and another that serves as regularizer. For example, let $\{(\mathbf{x}_t, \mathbf{y}_t)\}_{t=1}^T$ denote a training set of $T$ labeled examples, and let $\mathcal{C}(\mathbf{y}, \mathbf{f}(\mathbf{x}))$ denote the cost when a network's actual output $f(\mathbf{x})$ is evaluated against some reference output $\mathbf{y}$. We use

$$\mathcal{E}(\boldsymbol{\Theta}) = \frac{1}{T} \sum_t \mathcal{C}(\mathbf{y}_t, \mathbf{f}(\mathbf{x}_t)) \tag{12}$$

to denote the error obtained by averaging these costs over the training set. Likewise, we assume that the network's overall loss function is given by

$$\mathcal{L}(\boldsymbol{\Theta}) = \mathcal{E}(\boldsymbol{\Theta}) + \lambda \mathcal{R}(\mathbf{W}), \tag{13}$$

where $\mathcal{R}(\mathbf{W})$ is a regularizer that penalizes large weights[2] and the constant $\lambda \geq 0$ determines the amount of regularization. Note that while the error in eq. (12) may depend in a complicated way on the network's weights and biases, it is necessarily invariant to the rescaling transformations of these parameters in eqs. (4–5). On the other hand, typical regularizers are not invariant to rescaling transformations of the weights, and these are the sorts of regularizers that we consider here.

When a regularizer penalizes large weights, it is not only useful to prevent overfitting; in a feedforward network with rescaling symmetries, it also induces a natural way to measure if (and how well) the weights are balanced. Consider the effect of a rescaling transformation on the weights at any hidden unit $i \in \mathcal{H}$. A rescaling transformation with $a_i < 1$ redistributes the magnitudes of these weights in a forward direction (from incoming weights to outgoing weights), while a rescaling transformation with $a_i > 1$ will redistribute the magnitudes of weights in a backward direction (from outgoing to incoming weights). We formalize these ideas with the following definition so that we can speak interchangeably of *balancedness* and *stationarity*:

> **Definition 2** (Weight balancedness). *In a feedforward network with homogeneous activation functions, we say that the weights are balanced with respect to the regularizer $\mathcal{R}(\mathbf{W})$ if the regularizer is stationary with respect to infinitesimal instances of the rescaling transformations of the form in eq. (4).*

In section 2, we examined the minimality of the $\ell_{p,q}$-norm in eq. (1) with respect to rescaling transformations. The main ideas of this section, however, do not depend on that specific choice of norm, and therefore in what follows we state them in the more general terms of the above definition. To begin, we describe more precisely what it means for the regularizer $\mathcal{R}(\mathbf{W})$, or indeed any differentiable function of the weight matrix, to be stationary at some point with respect to infinitesimal rescaling transformations.

> **Lemma 3** (Stationarity). *Let $\mathcal{K}(\mathbf{W})$ be a differentiable function of the weight matrix in a feedforward network with homogeneous activation functions. Then $\mathcal{K}$ is stationary with respect to rescaling transformations at $\mathbf{W}$ if and only if*
>
> $$0 = \sum_j \left( W_{ij} \frac{\partial \mathcal{K}}{\partial W_{ij}} - W_{ji} \frac{\partial \mathcal{K}}{\partial W_{ji}} \right) \quad \text{for all} \quad i \in \mathcal{H}. \tag{14}$$

*Proof.* Consider a rescaling transformation of the form $a_i = 1 + \delta a_i$ for all $i \in \mathcal{H}$, where $\delta a_i$ denotes an infinitesimal variation. Under such a transformation, the weights of the network are rescaled via eq. (4) as

---

[2]The regularizer penalizes large weights, but not large biases, as only the former cause the outputs of the network to depend sensitively on the inputs to the network.

$W_{ij} \leftarrow W_{ij}(1 + \delta a_i)/(1 + \delta a_j)$; thus to lowest order, their change is given by $\delta W_{ij} = W_{ij}(\delta a_i - \delta a_j)$. Now we can work out (again to lowest order) the change $\delta \mathcal{K} = \mathcal{K}(\mathbf{W} + \delta \mathbf{W}) - \mathcal{K}(\mathbf{W})$ that results from this variation:

$$\delta \mathcal{K} = \sum_{ij} \frac{\partial \mathcal{K}}{\partial W_{ij}} \delta W_{ij} = \sum_{ij} \frac{\partial \mathcal{K}}{\partial W_{ij}} W_{ij}(\delta a_i - \delta a_j) = \sum_i \delta a_i \sum_j \left( \frac{\partial \mathcal{K}}{\partial W_{ij}} W_{ij} - \frac{\partial \mathcal{K}}{\partial W_{ji}} W_{ji} \right). \quad (15)$$

The lemma follows easily from this result. If eq. (14) is satisfied, then it is clear from eq. (15) that $\delta \mathcal{K} = 0$, proving one direction of implication; likewise if $\delta \mathcal{K} = 0$ for arbitrary infinitesimal variations $\delta a_i$, then each coefficient of $\delta a_i$ in eq. (15) must vanish, proving the other direction. $\qquad \square$

Regularizers were originally introduced to avoid overfitting (Goodfellow et al., 2016), but it is now widely appreciated that they also serve other purposes. It has been observed, for example, that regularizers can help to learn better models on the *training* data (Krizhevsky et al., 2012), suggesting that smaller weights in deep networks lead to better behaved gradients. Likewise, it has been observed that highly unbalanced weights lead to much slower training in ReLU networks; the reason is that partial derivatives such as $\partial L/\partial W_{ij}$ scale *inversely* as the weights under rescaling transformations (Neyshabur et al., 2015a; Dinh et al., 2017). More generally, it has been argued (van Laarhoven, 2017) that "by decreasing the scale of the weights, weight decay increases the effective learning rate" and that "if no regularization is used the weights can grow unbounded, and the effective learning rate goes to 0." All of the above observations suggest a role for learning algorithms that actively balance the weights of feedforward networks with homogeneous activation functions. Such algorithms can in turn be derived from the weight-balancing flows of the following theorem.

> **Theorem 4** (Weight-balancing flows). *Let $\mathcal{L}(\mathbf{\Theta})$ denote the regularized loss in eq. (13), and consider a network whose weights are initially balanced with respect to the regularizer $\mathcal{R}(\mathbf{W})$ and whose biases at hidden units are initialized at zero and never adapted. Then the weights will remain balanced with respect to $\mathcal{R}(\mathbf{W})$ if they evolve as*
>
> $$\frac{d}{dt}\left( W_{ij} \frac{\partial \mathcal{R}}{\partial W_{ij}} \right) = -W_{ij} \frac{\partial \mathcal{L}}{\partial W_{ij}}. \quad (16)$$

The theorem shows how to construct a canonical, weight-balancing flow for any differentiable regularizer. Before proving the theorem, we emphasize the precondition that the biases of all hidden (but not output) units are frozen at zero: i.e., $\dot{b}_i = b_i = 0$ for all $i \in \mathcal{H}$. As we shall see, this requirement[3] arises because we defined balancedness with respect to a regularizer $\mathcal{R}(\mathbf{W})$ that penalizes large weights *but not large biases*.

*Proof.* By Lemma 3, the weights are balanced with respect to the regularizer $\mathcal{R}(\mathbf{W})$ if they satisfy the stationarity conditions (substituting $\mathcal{R}$ for $\mathcal{K}$) in eq. (14). We prove the theorem by showing that the zeros in these stationarity conditions are conserved quantities analogous to those in eq. (11). As shorthand, let

$$Q_i = \sum_j \left( W_{ij} \frac{\partial \mathcal{R}}{\partial W_{ij}} - W_{ji} \frac{\partial \mathcal{R}}{\partial W_{ji}} \right) \quad (17)$$

denote the imbalance of incoming and outgoing weights at a particular hidden unit. To prove that $Q_i$ is conserved, we must show that $\dot{Q}_i = 0$ when the weights of the network evolve under the flow in eq. (16) from an initially balanced state. It follows from eq. (16) that

$$\frac{dQ_i}{dt} = -\sum_j \left( W_{ij} \frac{\partial \mathcal{L}}{\partial W_{ij}} - W_{ji} \frac{\partial \mathcal{L}}{\partial W_{ji}} \right). \quad (18)$$

From eq. (13) the network's loss function $\mathcal{L}(\mathbf{\Theta})$ consists of two terms, an empirical error $\mathcal{E}(\mathbf{\Theta})$ and a regularizer $\mathcal{R}(\mathbf{W})$. Substituting these terms into (18), we obtain

$$\frac{dQ_i}{dt} = -\sum_j \left( W_{ij} \frac{\partial \mathcal{E}}{\partial W_{ij}} - W_{ji} \frac{\partial \mathcal{E}}{\partial W_{ji}} \right) - \lambda \sum_j \left( W_{ij} \frac{\partial \mathcal{R}}{\partial W_{ij}} - W_{ji} \frac{\partial \mathcal{R}}{\partial W_{ji}} \right). \quad (19)$$

---

[3]There are other reasons besides weight-balancing to learn ReLU networks with zero biases. It has been noted that zero biases are necessary to learn *intensity-equivariant* representations of sensory inputs (Hinton et al., 2011; Mohan et al., 2020); these are representations in which the hidden-layer activations scale in proportion to the intensity of visual or auditory signals. Such networks also have certain margin-maximizing properties when they are trained by gradient flow (Lyu & Li, 2020).

Consider the left sum in eq. (19). If the hidden units do not have biases, then $\mathcal{E}(\boldsymbol{\Theta})$ is invariant under (and hence stationary with respect to) rescaling transformations of the weights *alone*, and the first term vanishes by Lemma 3 (substituting $\mathcal{E}$ for $\mathcal{K}$). Now consider the right sum in eq. (19); we see from eq. (17) that this term is proportional to $Q_i$ itself, so that

$$\frac{dQ_i}{dt} = -\lambda Q_i. \tag{20}$$

Finally, we observe that $Q_i = 0$ at time $t = 0$ if the weights are initially balanced. In this case, the only solution to eq. (20) is the trivial one with $Q_i = \dot{Q}_i = 0$ for all time, thus proving the theorem. $\qquad\square$

The calculation in this proof yields another insight. From eq. (20), we see that $Q_i$ decays exponentially to zero *if the weights are not initialized in a balanced state.* The decay is caused by the regularizer, an effect that Tanaka & Kunin (2021) describe as the Noether learning dynamics. However, the decay may be slow for the typically small values of the hyperparameter $\lambda > 0$ used in practice. Taken together, the fixes and flows in this paper can be viewed as a way of balancing the weights *at the outset and throughout the entire course of learning*, rather than relying on the asymptotic effects of regularization to do so in the limit.

A weight-balancing flow is only useful insofar as it also reduces the network's loss function. The weight-balancing flows in eq. (16) do not strictly follow the gradient of the loss function in eq. (13). Nevertheless, our final theorem shows that under relatively mild conditions these flows have the same property of descent.

---

**Theorem 5** (Balanced descent). *Let $\omega_{ij} = \log|W_{ij}|$. Then the weight-balancing flow in eq. (16) descends everywhere that the loss function is not stationary with respect to $\boldsymbol{\omega}$ and the regularizer has a positive definite Hessian with respect to $\boldsymbol{\omega}$:*

$$\frac{d\mathcal{L}}{dt} < 0 \quad wherever \quad \left|\frac{\partial \mathcal{L}}{\partial \boldsymbol{\omega}}\right| > 0 \quad and \quad \frac{\partial^2 \mathcal{R}}{\partial \boldsymbol{\omega} \partial \boldsymbol{\omega}^\top} \succ \mathbf{0}. \tag{21}$$

---

Before proving the theorem, we emphasize that *eq. (21) refers to the Hessian of the regularizer $\mathcal{R}(\mathbf{W})$, not the Hessian of the network's overall loss function $\mathcal{L}(\mathbf{W})$.* As we shall see, the former is very well behaved for typical regularizers, while the latter (about which one can say very little) depends in a complicated way on the network's fit to the training data.

*Proof.* The property of descent emerges for these flows in a similar way as for other generalized flows (Wibisono et al., 2016; Tanaka & Kunin, 2021). We begin by observing that the flow in eq. (16) takes a simpler form in terms of the variable $\boldsymbol{\omega}$. Since $\omega_{ij} = \log|W_{ij}|$, we have equivalently that $W_{ij}^2 = e^{2\omega_{ij}}$, and differentiating we find $\partial W_{ij}/\partial w_{ij} = W_{ij}$. We can use the chain rule to differentiate in eq. (16) with respect to $\boldsymbol{\omega}$ instead of $\mathbf{W}$. In this way we find (in vector notation) that

$$\frac{d}{dt}\left(\frac{\partial \mathcal{R}}{\partial \boldsymbol{\omega}}\right) = -\frac{\partial \mathcal{L}}{\partial \boldsymbol{\omega}}. \tag{22}$$

This equation specifies *implicitly* how the weights evolve in time through the form of the regularizer, $\mathcal{R}(\mathbf{W})$. To derive an explicit form for this evolution, we differentiate through the left side of the equation:

$$\frac{d}{dt}\left(\frac{\partial \mathcal{R}}{\partial \boldsymbol{\omega}}\right) = \frac{\partial^2 \mathcal{R}}{\partial \boldsymbol{\omega} \partial \boldsymbol{\omega}^\top} \cdot \frac{d\boldsymbol{\omega}}{dt}. \tag{23}$$

Note how the Hessian of the regularizer appears in this equation; as shorthand in what follows we denote this Hessian by $\mathbf{H}(\boldsymbol{\omega}) = \frac{\partial^2 \mathcal{R}}{\partial \boldsymbol{\omega} \partial \boldsymbol{\omega}^\top}$. Now suppose that $\mathbf{H}(\boldsymbol{\omega})$ is positive definite (hence also invertible) at the current values of the weights. Then combining eqs. (22–23), we see that

$$\frac{d\boldsymbol{\omega}}{dt} = -\mathbf{H}^{-1}(\boldsymbol{\omega}) \cdot \frac{\partial \mathcal{L}}{\partial \boldsymbol{\omega}}. \tag{24}$$

In sum, we have shown that if $\mathbf{H}(\boldsymbol{\omega})$ is positive definite, then the weight-balancing flow in eq. (16) has the same dynamics as eq. (24). Thus we can also interpret these dynamics as a generalized gradient flow in

which the weights are reparameterized in terms of $\omega_{ij} = \log|W_{ij}|$ and the gradient is preconditioned by the inverse Hessian $\mathbf{H}^{-1}(\boldsymbol{\omega})$. Now suppose further that the gradient $\partial\mathcal{L}/\partial\boldsymbol{\omega}$ does not vanish. Then we have

$$\frac{d\mathcal{L}}{dt} \;=\; \frac{\partial\mathcal{L}}{\partial\boldsymbol{\omega}}\cdot\frac{d\boldsymbol{\omega}}{dt} \;=\; -\frac{\partial\mathcal{L}}{\partial\boldsymbol{\omega}}\cdot\mathbf{H}^{-1}(\boldsymbol{\omega})\cdot\frac{\partial\mathcal{L}}{\partial\boldsymbol{\omega}} \;<\; 0. \tag{25}$$

This suffices to prove the theorem, but further intuition may be gained by comparing the argument in eq. (25) to the analogous one for gradient flow in eq. (10). The main differences are the appearance of the inverse Hessian preconditioner (via the regularizer) and the change of variables (from $W_{ij}$ to $\omega_{ij}$). $\qquad\square$

### 3.3 Flows for $\ell_{p,q}$-norm regularization

In the last section we derived weight-balancing flows for any regularizer $\mathcal{R}(\mathbf{W})$. The derivation was general, assuming only that the regularizer was differentiable. In this section we investigate the flow in eq. (16) for regularizers based on the $\ell_{p,q}$-norm in eq. (1). Specifically, we consider regularizers of the form

$$\mathcal{R}(\mathbf{W}) = \frac{1}{q}\sum_i\left(\sum_j |W_{ij}|^p\right)^{\frac{q}{p}}. \tag{26}$$

The flows induced by these regularizers have several interesting properties, which we briefly summarize. Proofs and further results can be found in appendix B.

To build intuition, we begin by noting some special cases of interest. The weight-balancing flow in eq. (16) reduces to familiar forms for the simplest choices of $p$ and $q$ in eq. (26). For example, when $p = q = 2$, this flow simplifies (for *nonzero* weights) to $\dot{W}_{ij} = -\frac{1}{2}\frac{\partial\mathcal{L}}{\partial W_{ij}}$, which can be approximated by gradient descent

$$W_{ij} \leftarrow W_{ij} - \eta\frac{\partial\mathcal{L}}{\partial W_{ij}} \tag{27}$$

with a small learning rate $\eta > 0$. On the other hand, when $p = q = 1$, the flow in eq. (16) simplifies to $\dot{W}_{ij} = -|W_{ij}|\frac{\partial\mathcal{L}}{\partial W_{ij}}$, which can be approximated by the *exponentiated* gradient descent

$$W_{ij} \leftarrow W_{ij}\exp\left(-\eta\cdot\text{sign}(W_{ij})\cdot\frac{\partial\mathcal{L}}{\partial W_{ij}}\right) \tag{28}$$

Similar updates based on exponentiated gradients have been studied in many different contexts (Kivinen & Warmuth, 1997; Arora et al., 2012; Bernstein et al., 2020). With these special cases in mind, we now state the main result of this section.

---

**Theorem 6** (Almost everywhere descent). *If the $\mathcal{R}(\mathbf{W})$ is based on the $\ell_{p,q}$-norm as in eq. (26), then the weight-balancing flow in eq. (16) decreases the regularized loss in every open orthant of weight space.*

---

Theorem 6 establishes the property of descent for weight-balancing flows based on $\ell_{p,q}$-norm regularization. The proof of this theorem requires two steps—first, to compute the Hessian for this regularizer, and second, to show that it is positive definite in every open orthant of the weight space. These steps are presented in appendix B.

## 4 Related work

The results in this paper build on earlier work. Most relevant to section 2 is the work on ENorm (Stock et al., 2019)—an elegant procedure, also based on multiplicative updates, to minimize the $\ell_p$-norm of the weights in ReLU networks. The updates for ENorm are derived from block coordinate descent of eq. (26) with respect to rescaling transformations for the special case $p = q$; the blocks are formed by grouping hidden units in the same layer. The updates in this paper are derived in a different way—from a so-called auxiliary function,

an approach used in algorithms for nonnegative matrix factorization (Lee & Seung, 2000) and Expectation-Maximization (Dempster et al., 1977). This approach (see appendix A) has two notable features. First, it leads to updates that are applied in parallel at all of the network's hidden units, and thus unlike those derived from block coordinate descent, they are not specialized to layered networks whose structure suggests a natural partition (i.e., blocking) of the hidden units. Second, it yields closed-form updates to minimize the $\ell_{p,q}$-norm despite the more complicated gradients that arise when $p \neq q$. This more general case is of interest because the *max-norm* emerges from eq. (1) in the limit $q \to \infty$ (Neyshabur et al., 2015a).

Our results in section 3 also build on previous work. Most relevant here is the work by Tanaka & Kunin (2021) that analyzes symmetry-breaking in deep learning via the continuous-time dynamics of damped Lagrangian systems. Their work gives an elegant and broadly unifying treatment for overparameterized networks with general symmetry groups. It builds in turn on the highly influential work of Wibisono et al. (2016), who introduced Bregman Lagrangians to analyze accelerated methods for optimization. Tanaka & Kunin (2021) stated that "a future direction is to extend our analysis to the case of rescale symmetry of the homogeneous activation functions such as ReLU." In this paper we have not made use of Bregman Lagrangians; instead we have given a proof of Theorem 4 stripped down to its bare essentials. To do so, we derived the flows of eq. (16) in a bottom-up fashion, starting from the requirement that balanced weights should remain balanced, rather than a top-down fashion, starting from a time-dependent Lagrangian and deriving its Euler-Lagrange equations. In the latter approach, the Bregman kinetic energy plays the role of the regularizer, breaking the rescaling symmetry of the Lagrangian and undoing the conservation law for the Noether charge $Q_i$ in eq. (17). We hope that some readers will appreciate the different path we have taken to obtain these results.

Many other studies have also investigated the relationship between rescaling transformations and learning. In a seminal paper, Neyshabur et al. (2015a) showed that stochastic gradient descent (SGD) performs poorly in highly unbalanced networks, and in its place, they proposed PathSGD, a rescaling-invariant procedure that approximates steepest descent with respect to a special path-based regularizer. Notably, this regularizer has the distinguishing property that it computes the minimum value of a max-norm regularizer, where the minimum is performed over all networks equivalent up to rescaling (Neyshabur et al., 2015b). PathSGD was followed by other formulations of rescaling-invariant learning. For example, Badrinarayanan et al. (2015) fixed the rescaling degrees of freedom by constraining weight vectors to have unit norm, while Meng et al. (2019) showed how to perform SGD in the vector space of paths (as opposed to weights), where the rescaling-invariant value of a path is given by the product of its weights. Also, Stock et al. (2019) trained networks by interweaving updates from SGD and ENorm with $\ell_2$-norm regularization; interestingly, the networks in these experiments generalized better on test data. This improvement suggests to interweave the more general updates in section 2, minimizing the $\ell_{p,q}$-norm, with learning rules based on their corresponding flows in section 3. Our study lays the foundation for this further exploration.

Even more recent work has suggested that learning can be accelerated by certain rescaling transformations. For instance, Armenta et al. (2021) showed that the magnitudes of backpropagated gradients in ReLU networks are increased on average by randomly rescaling their weights—a process they call *neural teleportation*. More generally, Zhao et al. (2022) explored how to choose symmetry group transformations that purposefully increase the norms of gradients for learning. Because these gradients are computed with respect to training examples, this approach can be viewed as a *data-driven* procedure for manipulating the optimization landscape via symmetry group transformations. Our approach differs from the above by aiming to minimize the norms of a network's weights rather than to maximize the norms of its gradients. We note that the latter are unbounded above with respect to the (non-compact) group of rescaling transformations, and therefore one must be careful to identify the regime in which they serve as a reliable proxy for rates of convergence.

## 5 Discussion

Many aspects of deep learning are not fully understood. In this paper we have shown that further understanding may be gained from the symmetries of multilayer networks and the analogies they suggest to physical systems. We have leveraged this understanding in two ways. First, we derived simple multiplicative updates that minimize the $\ell_{p,q}$-norm of the weight matrix over the equivalence class of networks related by

rescaling transformations. Second, we derived weight-balancing flows that preserve the minimality of any (differentiable) regularizer over the course of learning.

There are many questions deserving of further investigation. Most obviously, weight-balancing flows can be discretized to yield learning rules other than gradient descent. One important question is how to combine such learning rules with accelerated gradient-based methods, such as those involving momentum (Polyak, 1964) or adaptive learning schedules (Kingma & Ba, 2015; Duchi et al., 2010; Tieleman & Hinton, 2018), that are used in all large-scale experiments. Another is how weight-balancing relates to (or may be incorporated with) schemes such as batch normalization (Ioffe & Szegedy, 2015), weight normalization (Salimans & Kingma, 2016), and layer normalization (Ba et al., 2016). It will require a mix of empirical and theoretical investigation to understand the interplay of these methods (van Laarhoven, 2017). In this work, we have not relied heavily on the machinery of Bregman Lagrangians (Wibisono et al., 2016; Tanaka & Kunin, 2021), but it is clear that they provide a powerful framework for further progress.

Other potential benefits of weight-balancing are suggested in the more familiar setting of matrix factorization (Horn & Johnson, 2012). The basic problem of factorization is underdetermined: any matrix can be written in an infinite number of ways as the product of two or more other matrices. But there are certain canonical factorizations of large matrices, such as the singular value decomposition, that reveal a wealth of information (Eckart & Young, 1936). It is natural to ask whether the functions computed by multilayer networks can be represented in a similarly canonical way, and if so, whether such representations might suggest more effective strategies for pruning, compressing, or otherwise approximating their weight matrices.

Finally we note that there are many possible criteria for weight-balancing besides minimizing the $\ell_{p,q}$-norm of the weight matrix. It would be interesting to study other regularizers and their corresponding flows from Theorem 4. We believe that the present work can provide a template for these further investigations—and also that such investigations will reveal a similarly rich mathematical structure.

### Acknowledgements

The author is grateful to the reviewers and action editor (N. Cohen) for many helpful suggestions. He has also benefited from discussions of these ideas with many colleagues, including A. Barnett, A. Bietti, J. Bruna, M. Eickenberg, R. Gower, B. Larsen, E. Simoncelli, S. Villar, N. Wadia, A. Wibisono, A. Wilson, and R. Yu.

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

## A  Convergence of multiplicative updates

In this appendix we prove that the multiplicative updates in Algorithm 1 minimize the $\ell_{p,q}$ norm in eq. (1). The proof has many technicalities, so we begin by developing further intuitions in section A.1, which analyzes the fixed points of these updates, and section A.2, which relates the updates to those considered in previous work. Finally, section A.3 gives a formal proof of convergence.

### A.1  Analysis of fixed points

It is clear that Algorithm 1 can only converge to fixed points of the multiplicative updates. We begin with an elementary observation of where these fixed points occur.

**Proposition 7** (Fixed Points). *If a weight matrix $\mathbf{W}$ is a fixed point of the multiplicative updates in Algorithm 1, then all of the rescaling factors in eqs. (4,5,8) are equal to unity: i.e., $a_i = 1$ for all $i$.*

*Proof.* We proceed by induction. Note that $a_1 = 1$ because the network must have at least one input unit and the rescaling factors at all non-hidden units are fixed to one. Let $i > 1$ and suppose that $a_j = 1$ for all $j < i$. The $i^{\text{th}}$ unit in the network is either hidden ($i \in \mathcal{H}$) or not hidden ($i \notin \mathcal{H}$). If the latter, then as before we have trivially that $a_i = 1$. If the former, then there must exist some $j$ such that $W_{ij} \neq 0$ (since otherwise the unit would be unaffected by the network's inputs, a condition that we exclude). Note that the weight $W_{ij}$ is multiplicatively rescaled in eq. (4) by the ratio $a_i/a_j$. Since $W_{ij} \neq 0$, a fixed point occurs only when $a_i = a_j$, and since $a_j = 1$ by the inductive hypothesis, it follows that $a_i = 1$. $\qquad\square$

Next we consider what the fixed points of Algorithm 1 signify. Recall that the updates were introduced to minimize the norm in eq. (1). The next lemma shows that the fixed points correspond to global minima.

**Lemma 8** (Global Optimality). *If a weight matrix $\mathbf{W}$ is a fixed point of the multiplicative updates in Algorithm 1, then there exists no weight matrix $\mathbf{W}' \sim \mathbf{W}$ such that $\|\mathbf{W}'\|_{p,q} < \|\mathbf{W}\|_{p,q}$.*

*Proof.* Suppose that $\mathbf{W}' \sim \mathbf{W}$ with $W'_{ij} = W_{ij}(a_i/a_j)$ for some rescaling transformation $\mathbf{a} \in \mathcal{A}$, and consider how $\|\mathbf{W}'\|_{p,q}^q$ depends on the rescaling factors of this transformation. This dependence is captured (up to a multiplicative constant) by the continuous function $F : \mathcal{A} \to \Re$, where

$$F(\mathbf{a}) = \frac{1}{q} \sum_i \left( \sum_j |W_{ij}|^p (a_i/a_j)^p \right)^{\frac{q}{p}}. \tag{29}$$

It is instructive to examine the dependence of this function on the logarithms of the scaling factors, $\log a_i$, rather than the scaling factors themselves. Doing so, we find:

$$F(\mathbf{a}) = \frac{1}{q} \sum_i \exp\left( \frac{q}{p} \log \sum_j e^{p\left[ \log a_i - \log a_j + \log |W_{ij}| \right]} \right). \tag{30}$$

It follows from basic composition laws of convex functions (Boyd & Vandenberghe, 2004) that $\|\mathbf{W}'\|_{p,q}^q$ is itself a convex function of the vector $\log \mathbf{a} = (\log a_1, \log a_2, ..., \log a_n)$, and this in turn implies that *any stationary point of $F$ is a global minimizer of $F$*. To locate such a point, we examine where the partial derivatives of $F$ vanish. Starting from eq. (29), a tedious but straightforward calculation gives

$$\frac{\partial F}{\partial a_i} = \frac{1}{a_i} \left[ \left( \sum_j |W_{ij}|^p (a_i/a_j)^p \right)^{\frac{q}{p}} - \sum_j \left( \sum_k |W_{jk}|^p (a_j/a_k)^p \right)^{\frac{q}{p}-1} |W_{ji}|^p (a_j/a_i)^p \right]. \tag{31}$$

Now suppose that $\mathbf{W}$ is a fixed point of the multiplicative updates. Then by the previous proposition, it must be the case that the right hand side of eq. (8) is equal to unity. This occurs when the denominator and numerator of eq. (8) are equal, or equivalently when

$$\left( \sum_j |W_{ij}|^p \right)^{\frac{q}{p}} = \sum_j \left( \sum_k |W_{jk}|^p \right)^{\frac{q}{p}-1} |W_{ji}|^p. \tag{32}$$

Eq. (32) can be viewed as a balancing condition that holds at fixed points of the multiplicative updates: it equates a sum over incoming weights (on the left) with a sum over outgoing weights (on the right). Comparing the last two equations, we see another consequence of this balance; it implies that gradient of $F$ in eq. (31) vanishes when $\mathbf{a} = \mathbf{1}$, where $\mathbf{1} \in \Re^n$ is the vector of all ones. But this implies that a global minimum of $F(\mathbf{a})$ is obtained when $\mathbf{a}$ is the *identity* transformation. In this case, by definition there exists no $\mathbf{a}' \in \mathcal{A}$ such that $F(\mathbf{a}') < F(\mathbf{1})$, or equivalently, there exists no $\mathbf{W}' \sim \mathbf{W}$ such that $\|\mathbf{W}'\|_{p,q} < \|\mathbf{W}\|_{p,q}$. $\square$

To summarize, we have *not yet* shown that the updates in Algorithm 1 converge. But we have shown that if they *do* converge, they yield a functionally equivalent network whose weights minimize the norm in eq. (1).

## A.2 Relation to ENorm

Consider the special case of the norm in eq. (1) when $p=q$. This special case leads to many simplifications. When $p=q$, for example, the partial derivative in eq. (31) reduces to

$$\frac{\partial F}{\partial a_i} = \frac{1}{a_i} \left[ \sum_j |W_{ij}|^p (a_i/a_j)^p - \sum_j |W_{ji}|^p (a_j/a_i)^p \right]. \tag{33}$$

This expression is sufficiently simple to pursue a strategy of coordinate descent, minimizing $F(\mathbf{a})$ with respect to one rescaling factor at a time. In particular, consider the rescaling factor $a_i$ at some hidden unit ($i \in \mathcal{H}$), and suppose that $a_j = 1$ for all $j \neq i$. Then the partial derivative in eq. (33) vanishes when

$$a_i = \left[ \frac{\sum_j |W_{ji}|^p}{\sum_j |W_{ij}|^p} \right]^{\frac{1}{2p}}. \tag{34}$$

Eq. (34) is the basis for the ENorm algorithm (Stock et al., 2019), which uses multiplicative updates with rescaling factors, as in eqs. (4–5), to minimize the $\ell_{p,p}$-norm of the weight matrix. The updates for ENorm are derived from *block* coordinate descent of eq. (29) for the special case $p = q$; the blocks are composed of hidden units in the same layer of a deep network (for which the partial derivatives in eq. (33) decouple). As its name suggests, the procedure aims to equalize the $p$-norms of incoming and outgoing weights.

When $p = q$, the astute reader may have noticed a mismatch in the outer exponents of eqs. (8) and (34); the former is $\frac{1}{4p}$ for the multiplicative updates in this paper, while the latter is $\frac{1}{2p}$ for ENorm. The origin of this discrepancy, by a factor of two, will be further explained in section A.3. At a high level, the discrepancy arises because ENorm is based on block coordinate descent whereas the updates in this paper are applied in parallel at all the hidden units of the network. The larger exponent of ENorm corresponds to a more aggressive update, but one that is only applied to the rescaling factors in one layer of hidden units. The smaller exponent in our approach is needed to provide a similar guarantee of monotonic convergence when all the rescaling factors are updated in parallel. Compared to previous work, our results can be viewed as extending the reach of parallelizable multiplicative updates in two directions—first, to the larger family of feedforward (but not strictly layered) networks, and second, to the larger family of norms when $p \neq q$.

### A.3 Proof of convergence

Theorem 1 is proved by showing that the multiplicative updates in Algorithm 1 satisfy the preconditions for Meyer's convergence theorem (Meyer, 1976). Meyer's result itself builds on the convergence theory of Zangwill (1969). Our tools are similar to those that have been used to derive the convergence of other multiplicative updates with nonnegativity constraints (Lee & Seung, 1999; Saul et al., 2003; Sha et al., 2007) as well as more general iterative procedures in statistical learning (Dempster et al., 1977; Wu, 1983; Yuille & Rangarajan, 2003; Gunawardana & Byrne, 2005; Sriperumbudur & Lanckriet, 2012).

We prove the theorem with the aid of two additional lemmas. The first lemma considers the effect of a single rescaling transformation in the weight-balancing procedure of Algorithm 1. This lemma is at the heart of the algorithm: it shows that each multiplicative update reduces the entry-wise $\ell_{p,q}$-norm of the weight matrix.

---

**Lemma 9** (Monotone Improvement). *Suppose* $\mathbf{W}' \sim \mathbf{W}$*, and specifically, suppose* $W'_{ij} = W_{ij}(a_i/a_j)$ *where* $\mathbf{a} \in \mathcal{A}$ *is the vector of rescaling factors whose elements for* $i \in \mathcal{H}$ *are given by eq. (8). Then* $\|\mathbf{W}'\|_{p,q} \leq \|\mathbf{W}\|_{p,q}$*, and this inequality is strict unless* $\mathbf{W}' = \mathbf{W}$*.*

---

*Proof.* Recall the function $F : \mathcal{A} \rightarrow \Re$ from eq. (29), and recall in particular that $F(\mathbf{a}) = q^{-1}\|\mathbf{W}'\|_{p,q}^q$ if $W'_{ij} = W_{ij}(a_i/a_j)$. To prove the lemma, we must show equivalently that $F(\mathbf{a}) \leq F(\mathbf{1})$ when $\mathbf{a} \in \mathcal{A}$ is given by eq. (8) and also that this inequality is strict unless $\mathbf{a} = \mathbf{1}$. The proof is based on constructing a so-called auxiliary function, as in the derivations of the EM algorithm (Dempster et al., 1977), nonnegative matrix factorization (Lee & Seung, 2000), and the convex-concave procedure (Yuille & Rangarajan, 2003). Specifically, we seek an auxiliary function $G : \mathcal{A} \rightarrow \Re$ with the following three properties:

   (i) $F(\mathbf{a}) \leq G(\mathbf{a})$ for all $\mathbf{a} \in \mathcal{A}$.
   (ii) $F(\mathbf{1}) = G(\mathbf{1})$ where $\mathbf{1}$ is the vector of all ones.
   (iii) $G : \mathcal{A} \rightarrow \Re$ has a unique global minimizer at $\mathbf{a} \in \mathcal{A}$ given by eq. (8).

Suppose, for instance, that we can construct a function with these properties. Then at the minimizer $\mathbf{a} \in \mathcal{A}$ given by eq. (8), we have at once that

$$F(\mathbf{a}) \leq G(\mathbf{a}) \leq G(\mathbf{1}) = F(\mathbf{1}), \tag{35}$$

where the inequalities in eq. (35) follow from properties (i) and (iii) of the auxiliary function and the equality follows from property (ii). This proves the first part of the lemma. Now if $\mathbf{a} \neq \mathbf{1}$ in eqs. (8) and (35), then $G(\mathbf{a}) < G(\mathbf{1})$ (because $G$ has a *unique* global minimizer) and by extension $F(\mathbf{a}) < F(\mathbf{1})$. On the other hand, if $\mathbf{a} = \mathbf{1}$, then $\mathbf{W}$ is a fixed point of the multiplicative updates, and this proves the second part of the lemma.

The more challenging part of the proof is to *construct* an auxiliary function with these properties. As shorthand, let $r = \max(p, q)$, and consider the function

$$G(\mathbf{a}) \;=\; \tfrac{1}{2r} \sum_{ij} \rho_i(\mathbf{W})\, \pi_{ij}(\mathbf{W}) \left( a_i^{2r} + a_j^{-2r} \right) \;+\; \max\left(0, 1 - \tfrac{q}{p}\right) F(\mathbf{1}). \tag{36}$$

We claim that eq. (36) satisfies the three properties listed above, and we verify them in order of difficulty. First, we verify property (ii): comparing eqs. (29) and (36), we see by simple substitution that $F(\mathbf{1}) = G(\mathbf{1})$. Next we verify property (iii). We begin by observing that $\sum_j |W_{ij}|^p > 0$ and $\sum_j |W_{ji}|^p > 0$ for all $i \in \mathcal{H}$; these conditions are necessary (as mentioned in section 2.1) to ensure that each hidden unit lies on some directed path from the network's inputs to outputs. It follows that the coefficients of $a_i^{2r}$ and $a_j^{-2r}$ in eq. (36) are strictly positive, and hence $G(\mathbf{a})$ is *strongly convex* in the variable $(a_1^r, a_2^r, \ldots, a_n^r)$. By setting $\partial G/\partial a_i^r = 0$ for all $i \in \mathcal{H}$, we obtain the solution in eq. (8), and by strong convexity this solution must correspond to a unique global minimizer. This verifies property (iii) of the auxiliary function, and it also accounts for the peculiar exponent of $\frac{1}{4r}$ where $r = \max(p, q)$, that appears in the multiplicative update.

Finally, we verify property (i) of the auxiliary function. To do this, we must work out separately the cases for $p < q$ and $p > q$. These regimes require different arguments to establish the lower-bounding property of the auxiliary function. We also remind the reader of the definitions in eqs. (6–7).

- **Case 1:** $p \leq q$. To verify property (i) in this case we must show that $F(\mathbf{a}) \leq G(\mathbf{a})$ when $p \leq q$. Starting from eq. (29) for $F(\mathbf{a})$, we can derive the auxiliary function in eq. (36) via two simple inequalities:

$$F(\mathbf{a}) \;=\; \tfrac{1}{q} \sum_i \left( \sum_j |W_{ij}|^p (a_i/a_j)^p \right)^{\frac{q}{p}}, \tag{37}$$

$$=\; \tfrac{1}{q} \sum_i \rho_i(\mathbf{W}) \left( \sum_j \pi_{ij}(\mathbf{W})\, (a_i/a_j)^p \right)^{\frac{q}{p}}, \tag{38}$$

$$\leq\; \tfrac{1}{q} \sum_i \rho_i(\mathbf{W}) \sum_j \pi_{ij}(\mathbf{W}) (a_i/a_j)^q, \tag{39}$$

$$\leq\; \tfrac{1}{2q} \sum_{ij} \rho_i(\mathbf{W})\, \pi_{ij}(\mathbf{W}) \left( a_i^{2q} + a_j^{-2q} \right), \tag{40}$$

$$=\; G(\mathbf{a}) \quad \text{for} \quad p \leq q. \tag{41}$$

In eq. (39) we have used Jensen's inequality, exploiting the fact that $\pi_{ij}$ is a stochastic matrix, and in eq. (40) we have appealed to the inequality of arithmetic and geometric means. Finally, eq. (41) follows upon resubstituting eq. (7) into the previous line. This verifies property (i) for the case $p \leq q$.

- **Case 2:** $p \geq q$. Now we must show that $F(\mathbf{a}) \leq G(\mathbf{a})$ when $p \geq q$. Here we exploit the fact that a concave function (e.g., $x^{\frac{q}{p}}$) lies below any one of its tangents. It follows that

$$F(\mathbf{a}) \;=\; \tfrac{1}{q} \sum_i \left( \sum_j |W_{ij}|^p (a_i/a_j)^p \right)^{\frac{q}{p}}, \tag{42}$$

$$\leq\; F(\mathbf{1}) + \tfrac{1}{p} \sum_{ij} \left( \sum_k |W_{ik}|^p \right)^{\frac{q}{p}-1} |W_{ij}|^p \left( (a_i/a_j)^p - 1 \right), \tag{43}$$

$$=\; F(\mathbf{1}) + \tfrac{1}{p} \sum_{ij} \rho_i(\mathbf{W})\, \pi_{ij}(\mathbf{W}) \left( (a_i/a_j)^p - 1 \right), \tag{44}$$

$$\leq\; F(\mathbf{1}) \left( 1 - \tfrac{q}{p} \right) + \tfrac{1}{2p} \sum_{ij} \rho_i(\mathbf{W})\, \pi_{ij}(\mathbf{W}) \left( a_i^{2p} + a_j^{-2p} \right) \tag{45}$$

$$=\; G(\mathbf{a}) \quad \text{for} \quad p \geq q. \tag{46}$$

In sum, the auxiliary function $G(\mathbf{a})$ is derived by identifying those parts of $\|\mathbf{W}'\|_{p,q}^q$ that can be bounded by elementary inequalities. In doing so, we find that different bounds are required for the cases $p < q$ and $p > q$, and these differences account for the exponent $r = \max(p, q)$ that appears in the multiplicative updates. $\quad\square$

Lemma 9 shows that each multiplicative update in Algorithm 1 decreases $\|\mathbf{W}\|_{p,q}$ unless $\mathbf{W}$ is itself a fixed point. The lemma also rules out oscillations between distinct global minima. But this by itself is not enough to prove that the updates converge. To do this, we must also show that none of the rescaling factors increase without bound. This is the content of the next lemma.

**Lemma 10** (Compactness of Sublevel Sets). *Let $\mathcal{C} = \{\mathbf{W}' \,|\, \mathbf{W}' \sim \mathbf{W}, \|\mathbf{W}'\|_{p,q} \leq \|\mathbf{W}\|_{p,q}\}$ denote the sublevel set of weight matrices whose $\ell_{p,q}$-norm does not exceed that of $\mathbf{W}$. This set is compact.*

*Proof.* Let $F : \mathcal{A} \to \Re$ be the function, as defined in eq. (29), that computes the change in the $\ell_{p,q}$-norm of the weight matrix after a rescaling transformation. To prove the lemma, we must show equivalently that the sublevel set given by $\mathcal{F_1} = \{\mathbf{a} \in \mathcal{A} \,|\, F(\mathbf{a}) \leq F(\mathbf{1})\}$ is compact. It follows from the continuity of $F$ that its sublevel sets are closed; thus it remains only to show that $\mathcal{F_1}$ is bounded. At a high level, this boundedness will follow from the fact that the network has finite depth; a similar result has been obtained for the special case $p = q$ (Stock et al., 2019). In particular, suppose $\mathbf{a} \in \mathcal{F_1}$ with $F(\mathbf{a}) \leq F(\mathbf{1})$. Then if $W_{ij} \neq 0$, it must be the case that

$$\frac{a_i}{a_j} \leq \frac{(qF(\mathbf{1}))^{\frac{1}{q}}}{|W_{ij}|}, \tag{47}$$

because otherwise the $ij^{\text{th}}$ term of the sum in eq. (29) would by itself exceed $F(\mathbf{1})$. Let $j_0 \to j_1 \to \cdots \to j_m$ denote an $m$-step path through the network that starts at some input unit ($j_0 \in \mathcal{I}$), passes through the $i^{\text{th}}$ hidden unit after $k$ steps (so that $j_k = i$), ends at some output unit ($j_m \in \mathcal{O}$), and traverses only nonzero weights $W_{j_{\ell-1} j_\ell} \neq 0$ in the process. Note that there must exist at least one such path if the $i^{\text{th}}$ hidden unit contributes in some way to the function computed by the network. Since $a_{j_0} = 1$ and $a_{j_k} = a_i$, it follows that

$$a_i = \frac{a_{j_k}}{a_{j_0}} = \prod_{\ell=1}^{k} \frac{a_{j_\ell}}{a_{j_{\ell-1}}} \leq \prod_{\ell=1}^{k} \frac{(qF(\mathbf{1}))^{\frac{1}{q}}}{|W_{j_\ell j_{\ell-1}}|}, \tag{48}$$

where the inequality follows from eq. (47). Likewise, since $a_{j_m} = 1$ and $a_{j_k} = a_i$, it follows that

$$\frac{1}{a_i} = \frac{a_{j_m}}{a_{j_k}} = \prod_{\ell=k+1}^{m} \frac{a_{j_\ell}}{a_{j_{\ell-1}}} \leq \prod_{\ell=k+1}^{m} \frac{(qF(\mathbf{1}))^{\frac{1}{q}}}{|W_{j_\ell j_{\ell-1}}|} \tag{49}$$

Eqs. (48–49) provide upper and lower bounds on $a_i$ for all $\mathbf{a} \in \mathcal{F_1}$. Thus $\mathcal{F}_1$ is closed and bounded, hence compact. $\quad\square$

The previous two lemmas establish the necessary conditions for Meyer's monotone convergence theorem (Meyer, 1976). Armed with these lemmas, we can now prove Theorem 1.

*Proof of Theorem 1.* Let $\mathbf{W}_0$ denote an initial weight matrix. From Lemma 10, it follows that the set $\mathcal{C} = \{\mathbf{W} \,|\, \mathbf{W} \sim \mathbf{W}_0, \|\mathbf{W}\|_{p,q} \leq \|\mathbf{W}_0\|_{p,q}\}$ is compact. From Lemma 9, it follows that each multiplicative update yields a weight matrix $\mathbf{W} \in \mathcal{C}$ whose $\ell_{p,q}$-norm is less than or equal to the previous one (with equality occurring only when $\mathbf{W}$ is both a global minimizer and a fixed point of the updates). Finally, from eq. (8) we note that the multiplicative coefficients are a continuous function of the weights from which they are derived. The procedure in Algorithm 1 therefore satisfies the preconditions of compactness, strict monotonicity, and continuity for Meyer's monotone convergence theorem (Meyer, 1976) in the setting where fixed points occur at (and only at) global minima of $\|\mathbf{W}\|_{p,q}$ in $\mathcal{C}$. $\quad\square$

# B   Weight-balancing flow for $\ell_{p,q}$-norm regularization

In this appendix we prove Theorem 6 that the weight-balancing flow for the regularizer $\mathcal{R}(\mathbf{W})$ in eq. (26) descends in the network's regularized loss function. As a preliminary step, we remind the reader of the per-unit regularizer $\rho_i(\mathbf{W})$ defined in eq. (6) and the stochastic matrix with elements $\pi_{ij}(\mathbf{W})$ defined in eq. (7). From these definitions, it is also a straightforward exercise to verify that $W_{ij}\frac{\partial \mathcal{R}}{\partial W_{ij}} = \pi_{ij}(\mathbf{W})\rho_i(\mathbf{W})$, and this identity is useful in what follows.

*Proof of Theorem 6.* Consider the regularizer $\mathcal{R}(\mathbf{W})$ in eq. (26) as a function of the variable $\omega_{ij} = \log|W_{ij}|$. To prove the result, we show that the Hessian $\mathbf{H}(\boldsymbol{\omega}) = \frac{\partial^2 \mathcal{R}}{\partial\boldsymbol{\omega}\partial\boldsymbol{\omega}^\top}$ in Theorem 5 is positive definite whenever no weight $W_{ij}$ is equal to zero, or equivalently when every $\omega_{ij}$ is finite. The proof requires two steps—first, to compute the Hessian, and second, to show that it is positive definite. For the first step, we begin by evaluating the time-derivative in the flow of eq. (16). Here we find, by repeated differentiation, that

$$\frac{d}{dt}\left[W_{ij}\frac{\partial \mathcal{R}}{\partial W_{ij}}\right] = \frac{d}{dt}\Big[\pi_{ij}(\mathbf{W})\rho_i(\mathbf{W})\Big], \tag{50}$$

$$= \left[\sum_k \frac{\partial \pi_{ij}}{\partial W_{ik}}\dot{W}_{ik}\right]\rho_i(\mathbf{W}) + \pi_{ij}(\mathbf{W})\left[\sum_k \frac{\partial \rho_i}{\partial W_{ik}}\dot{W}_{ik}\right], \tag{51}$$

$$= p\pi_{ij}(\mathbf{W})\sum_k \Big[\delta_{jk} - \pi_{ik}(\mathbf{W})\Big]\frac{\dot{W}_{ik}}{W_{ik}}\rho_i(\mathbf{W}) + q\pi_{ij}(\mathbf{W})\sum_k \Big[\pi_{ik}(\mathbf{W})\rho_i(\mathbf{W})\Big]\frac{\dot{W}_{ik}}{W_{ik}} \tag{52}$$

$$= \pi_{ij}(\mathbf{W})\rho_i(\mathbf{W})\Big[p\,\dot{\omega}_{ij} + (q{-}p)\sum_k \pi_{ik}(\mathbf{W})\,\dot{\omega}_{ik}\Big]. \tag{53}$$

We can now read off the nonzero elements of $\mathbf{H}(\boldsymbol{\omega})$ from the relation between eqs. (23) and (53). Let $\mathbf{V}$ be any nonzero matrix of the same size as $\mathbf{W}$, and let $\mathtt{Vec}(\mathbf{V})$ denote the flattened representation of $\mathbf{V}$ as a vector. Then from eqs. (23) and (53) we have

$$\mathtt{Vec}(\mathbf{V})^\top \mathbf{H}(\boldsymbol{\omega})\,\mathtt{Vec}(\mathbf{V})$$
$$= p\sum_{ij}\pi_{ij}(\mathbf{W})\rho_i(\mathbf{W})\,V_{ij}^2 + (q{-}p)\sum_{ijk}\pi_{ij}(\mathbf{W})\rho_i(\mathbf{W})\pi_{ik}(\mathbf{W})\,V_{ij}V_{ik}, \tag{54}$$

$$= p\sum_{ij}\rho_i(\mathbf{W})\pi_{ij}(\mathbf{W})\left(V_{ij} - \sum_k \pi_{ik}(\mathbf{W})\,V_{ik}\right)^2 + q\sum_i \rho_i(\mathbf{W})\left(\sum_j \pi_{ij}(\mathbf{W})\,V_{ij}\right)^2, \tag{55}$$

$$> 0 \quad \text{for all} \quad \mathbf{V}\neq\mathbf{0}. \tag{56}$$

The strict inequality in the last line is justified by the following observations: (i) both $\pi_{ij}(\mathbf{W})$ and $\rho_i(\mathbf{W})$ are strictly positive when no weights are exactly equal to zero; (ii) the left term in eq. (55) only vanishes when $\mathbf{V}$ has constant rows; (iii) the right term only vanishes when for all rows $\sum_j \pi_{ij}(\mathbf{W})V_{ij} = 0$. Since $\mathbf{V}$ is nonzero, however, it is not possible to satisfy both of these conditions simultaneously, so that one term and/or the other must be strictly positive. Finally, since $\mathtt{Vec}(\mathbf{V})^\top\mathbf{H}(\boldsymbol{\omega})\,\mathtt{Vec}(\mathbf{V}) > 0$ for any nonzero $\mathbf{V}$, we conclude that $\mathbf{H}(\boldsymbol{\omega})$ is positive definite in every open orthant of the weight space. The result then follows from Theorem 5. $\qquad\square$

