# OpenReview forum: "Weight-balancing fixes and flows for deep learning"
_TMLR — Accepted by TMLR_

### Review · Reviewer_h4c9 · 2023-07-18

**Summary Of Contributions:**

This paper studies the rescaling invariance of feed-forward neural networks. Indeed, for instance, for ReLU non-linearity, one can obtain a functionally equivalent configuration of weights by taking a single neuron and multiplying all the input weights by some scalar $a$ and dividing all the output weights by $a$ (see Fig. 1 of the paper). The authors break this invariance by introducing regularization and looking for the weights with the minimal regularizer (norm) value not changing the outputs of the network.

The main contributions of the paper are the algorithm that finds the configuration of weights with the minimal norm preserving the functional form (Algorithm 1 and Theorem 2.1), and the continuous-time dynamic that minimizes the loss for some class of regularizers while preserving the “minimal regularizer value” property on the way (Theorems 3.3 and 3.4).

In more detail, the proposed Algorithm 1 operates as follows. For the given configuration of parameters, it iteratively rescales the weights and biases of fully connected layers preserving the functional form (i.e. the outputs for all possible inputs) and minimizing the total norm of the weights. The authors then prove that the algorithm converges and also demonstrate it empirically for several randomly initialized MLPs (multilayer perceptrons). Thus, Algorithm 1 allows fixing the gauge, i.e. choosing the symmetry with the minimal norm.

In Theorems 3.3 and 3.4, the authors propose a continuous time dynamic of weights that minimizes the loss and converges to the minimal gauge in terms of the regularizer. In particular, if the gauge is chosen to be optimal from the beginning (e.g. via applying Algorithm 1 to the randomly initialized network) it will remain optimal throughout the “training”. However, the authors do not discuss immediate practical benefits and do not study the proposed dynamics empirically.


**Audience:**

Yes

**Broader Impact Concerns:**

The paper does not bring broad impact concerns.

**Claims And Evidence:**

Yes

**Requested Changes:**

To summarize, the presented study looks anticlimactic. As mentioned before, the two main contributions are independent of each other. Furthermore, each of them does not provide an immediate application or solve a theoretical problem.

Throughout the paper, the authors discuss potential applications of their work (e.g. setting $q \to \infty$ in the norm). I believe that the paper would benefit from an empirical study of these discussed applications. For instance, one might demonstrate the benefits of the proposed dynamic (in Section 3) where the initial gauge is fixed using the algorithm from Section 2. This would both tie the parts of the paper and motivate the developments.

**Strengths And Weaknesses:**

### Strengths

- The authors do a good job of motivating the proposed developments with relevant literature, i.e. the approached problems seem actual and the proposed solutions relevant.
- To the best of my knowledge, the presented results are novel.
- Besides some minor concerns, the presentation is very clear.

### Weaknesses

In the current state, the paper seems like two independent studies both of which are incomplete. Indeed, each of the two presented results can be proposed separately without any loss of content. This by itself is a major concern indicating that the paper requires a major revision.

Furthermore, the developments of the first part of the paper (Section 2) are incremental compared to [2]. Indeed, the main development of this section can be summarized as an extension of the result of [2] from $l_{p,p}$ norms to $l_{p,q}$ norms. Also, [2] provides an extensive empirical study demonstrating the practical benefits of the proposed algorithm, while the practical benefits of the proposed extension are not demonstrated.

The benefits of the second part (Section 3) are unclear. The authors briefly discuss potential discretization schemes of the proposed dynamics, but these discussions are limited to future work discussion, without any empirical studies of the proposed schemes. From the theoretical perspective, there are no immediate implications, e.g. no novel analysis tools are proposed (as done in [1]).

Other comments:

- The last paragraph of Section 1 (page 2). It is not clear what the “speed” of the network means, as well as the “eventual outcomes of learning”.
- Formulation of Theorem 2.1. I suggest mentioning that the updates of Algorithm 1 preserve the functional form. In general, it is clear from the previous discussion, but it would improve the clarity of the statement, which is the main result.
- Proof of Proposition 2.2. Typo on the second line of the proof.
- Page 14. Typo in the first row. Typo in the first row of the second paragraph.
- The role of Definition 3.1 is not clear. Indeed, it seems like it’s simply a change of the word “stationary” for “balanced”.
- Theorem 3.3. The formulation “biases are frozen” is not suitable for a theorem statement.
- Page 21. It is not clear what “boosted gradients” mean.

[1] Tanaka, H., & Kunin, D. (2021). Noether’s learning dynamics: Role of symmetry breaking in neural networks. *Advances in Neural Information Processing Systems*, *34*, 25646-25660.
[2] Stock, P., Graham, B., Gribonval, R., & Jégou, H. (2019). Equi-normalization of neural networks. *arXiv preprint arXiv:1902.10416*.

---

> ### Author Response · Authors · 2023-07-24
> **Reply to reviewer h4c9**
>
> Thanks very much for your time and consideration of this manuscript.
>
> * You state that the developments of the first part of the paper are incremental, only extending a previous result of $\ell_{p,p}$ norms to $\ell_{p,q}$ norms. It is hopefully clear that this extension required an entirely different methodology; in particular, the updates for the $\ell_{p,p}$-norm can be derived simply by computing gradients and setting them equal to zero, whereas the updates in this paper require the additional (non-trivial) machinery of auxiliary functions. In addition, these updates can be applied in parallel at all hidden units rather than in a strictly layerwise fashion.
>
> * You state that that benefits of the second part are unclear because there are no empirical evaluations. We are also interested in further empirical evaluations, but it is clear that such evaluations will require an entirely different type of expertise (e.g., in more the practical aspects of deep learning) than what was required to prove the theoretical results in this paper. These empirical evaluations will require a study of many further issues, including how to best discretize the flows in eq. (2), how to combine them with other schemes for learning (e.g., momentum), and how to numerically implement the limit $q\rightarrow\infty$ in a stable manner. These are not small issues. The more modest goal of the paper is to lay the theoretical foundations for such work, and  especially to make the weight-balancing flows in eq. (2) intelligible to a large audience.
>
> * You state that the second part does not provide any novel tools for analysis. We believe that many will be intrigued with the particular manner in which we derive the weight-balancing flows in this section, and also that our methods can be more directly applied to study other classes of regularizers.
>
> * You expressed that the two parts of this paper are independent and not closely linked. The second part of the paper would be of limited interest, in our view, if one could not demonstrate that the weights could be initially balanced for a large class of regularizers (Algorithm 1). Likewise, the first part of the paper would be of limited interest if one could not also derive the specific form of the weight-balancing flows for this class of regularizers (Corollary 3.7).
>
> Minor comments:
>
> * On page 1, by "speed" we mean the rate at which the network converges to a solution of small training error, and by "outcomes of learning" we mean the types of solutions that are learned and how well they generalize. We can clarify this in a revised manuscript.
>
> * We agree in the formulation of Theorem 2.1 that it would add clarity to mention that the updates of Algorithm 1 preserve the functional form of the network.
>
> * We will fix the typos.
>
> * Definition 3.1 is, as you suggest, intended to link the ideas of stationarity and balancedness. It seems necessary to us to make this connection as explicit as possible. We can discuss this intention in the text to clarify the role of the definition (which is simply to highlight this connection).
>
> * Theorem 3.3: we can change "frozen biases" to "biases that are initialized at zero and never adapted."
>
> * Page 21: "boosted" here simply means "increased" (and we agree that the latter is more clear).
>
> Thanks again for the many suggestions and constructive feedback on this manuscript. We hope you agree that the weight-balancing flows in eq (2) have a surprisingly simple form, that we have shown how to derive these flows in a novel and highly intelligible manner, and that they deserve to be further studied and more widely known.

---

### Review · Reviewer_zCJg · 2023-07-20

**Summary Of Contributions:**

The paper studies gauge symmetries of deep feedforward neural networks with positively homogenous activation functions. First the authors propose an algorithm for finding the minimum $\ell_{p,q}$ norm weight matrix while preserving the network function by iteratively rescaling the incoming and outgoing weights. The algorithm enjoys convergence guarantees as shown in the paper. Secondly, the authors propose a flow that preserves the regularization term $\mathcal{R}(W)$ which is suggested to be minimized initially at random initialization. The proposed gauge-fixing flow also decreases the loss as proven in the paper, hence it is a valid flow.

**Audience:**

Yes

**Claims And Evidence:**

Yes

**Requested Changes:**

I am not sure if gauge-symmetries are a property of overparameterization as suggested by the very first paragraph of the paper. I'd suggest removing this.

I got confused for some time about the unusual indexing of hidden neurons. The weight matrix is in fact not only lower-triangular but also even in the lower-triangular entries there are many zeros (for example, $[d] \times [d]$ are not linked as the input nodes are not connected). It would be good to clarify this early on.

Question to authors: would it not be better to use $ W' \leftarrow^a W $ instead of $ W' \sim^a W $ ? The first implies that $W' $ comes after updating $W$ whereas the second feels like the order does not matter.

**Minor feedback/questions**

Theorem 2.1, I believe argmin should not include biases $b$.

Proposition 2.2 [proof], how is it converse statement trivial? Could the authors elaborate on this please?

Lemma 2.4 [proof, (iii)], I believe $G(1)$ is maximizer here. Also below Eq. (18).


**Strengths And Weaknesses:**

**Strengths:**
* The paper is pleasant to read.
* The results of this paper are novel to the best of my knowledge.
(Weight-balancing algorithm and the gauge-fixing flow in particular)
* The paper is theoretically well grounded, and how it is different than the relevant work is well explained.
* The norm minimization algorithm is shown to work well numerically for deep networks at initialization.

As the authors noted, the proposed flow converges to $Q_i \to 0$ exponentially even if $Q_i$'s are not initialized at zero.
It is true that this convergence can take a long time for small $\lambda$, whereas the norm minimization algorithm is converging quickly
in numerics. Overall the paper proposes a novel initialization (random init + weight-balancing algorithm) and a novel training paradigm with theoretical guarantees. At convergence, the weights that are initialized balanced would remain balanced under gradient flow as we know from the literature. This paper proposes another flow that preserves in addition the minimum $\ell_{p,q}$ norm. This might bring better inductive biases than gradient flow and better generalization guarantess. For the latter case, as the authors noted, an interesting case is the limit $q \to \infty$.

**Weaknesses:**
* The paper is too long and unfortunately at times repetetive. I think the first paragraph of Section 3 was discussed before, again on page 12, we are reminded that $\| \|_{p,q} $denotes the \ell_{p,q}$-norm, at the top of page 16, there is yet a similar related work discussion for the line of work around building on Noether's theorems for deep learning, and again at the end there is a further related works section.
The discussion also reads somehow verbose (I do not see how the third paragraph is related).
* Section 2.4 is relation to previous work once again. It is helpful to see how the formula simplifies for the special case $p=q$, however the text therein reads repetetive. Bottom of page 13 is again comparison to related work! It would really help to reader to have one related works section at the beginning (or at the end), and limit the discussion elsewhere that are specific to theorems to one-two sentences.
* The paper would benefit greatly from numerical experiments, similar to Figure 1 that compares gradient flow vs gauge-flow on the regularized objective for some simple tasks. However, this work builds up the theoretical foundation and I don't think these experiments are necessary for publication at TMLR.

---

> ### Author Response · Authors · 2023-07-24
> **Reply to Reviewer zCJg**
>
> Thanks for your time and consideration of this manuscript. We appreciate your many constructive suggestions for improvement.
>
> * We can submit a revised manuscript that is less repetitive in the ways that you suggest. The original manuscript perhaps erred on the side of repetition for the sake of clarity and in an attempt not to slight the authors of previous work. We can follow your suggestions to reduce the overall length.
>
> * We are also very interested in an empirical investigation of weight-balancing flows. Suffice it to say, this investigation will require not only an extensive amount of additional work, but also a very different type of expertise than what was needed for the results in this paper. By publishing the present results in TMLR, our goal is to attract collaborators for this effort.
>
> * You ask if gauge symmetries are a property of overparameterization. We agree that not all overparameterized models have gauge symmetries, and we did not mean to imply this (though we can see that the current text does carry this implication). We will reword this text more carefully, along the following lines: "in some overparameterized models it may be possible to identify internal (or gauge) symmetries under which the model's predictions are invariant." Would that suffice?
>
> * We can clarify that not only is the weight matrix lower-diagonal, but also that many of its lower-triangular entries are zero (e.g., between input units). We can see why this was a source of confusion.
>
> * We agree that there is an unfortunate ambiguity in ordering that is created by the notation $W'\sim^a W$.  This shorthand is not used very often, so perhaps we can simply write explicitly that $W'\sim W$ with $W^\prime_{ij} =W_{ij} (a_i/a_j)$ for some $a$. (We like to preserve the equivalence operator and the notion of equivalence classes.)
>
> * You suggest that the argmin in Theorem 2.1 should not include the biases $b$. But if they were dropped from the argmin, then the rescaled biases would be an unspecified variable on the right side of the optimization. Our understanding is that in constrained optimizations, the variables in constraints appear in the list of optimized variables even if they do not appear in the loss function itself: e.g., $\min_{x,y} (x^2)$ such that $x+y=1$ and $x^2+y^2\leq 2$.
>
> * Proposition 2.2: the proposition states that if $a$ is a fixed point of the weight matrix under the multiplicative updates of Algorithm 1, then $a_i=1$ for all units in the network. (The proposition is needed to rule out other types of fixed points.) By the trivial converse, we meant that if $a_i=1$ for all units in the network, then the weight matrix is clearly fixed under the multiplicative update $W_{ij} \leftarrow W_{ij} (a_i/a_j)$. We can reword this to be less confusing (or simply eliminate this observation).
>
> * Lemma 2.4 [proof (iii) and eq. 18]: can you explain why you believe $G$ should be maximized here? The auxiliary function $G$ in eq. (19) is convex, and it is being minimized by the algorithm's multiplicative update. (In other settings, such as the EM algorithm for maximum likelihood estimation, the auxiliary function is concave and maximized, but here it is the other way around.)
>
> Thanks again for your comments.

---

### Review · Reviewer_LFZ9 · 2023-07-23

**Summary Of Contributions:**

This paper makes two contributions to our understanding of feedforward neural networks with homogeneous activation functions. The first is a procedure for "gauge-fixing" - computing multiplicative rescaling factors at each hidden unit that rebalance the weights of these networks without altering the functions they compute. This is achieved by determining the functionally equivalent network whose weight matrix is of minimal $l_{p,q}$-norm. The paper provides an algorithm to compute these multiplicative weight updates which is also a contribution. Notably, these rescaling factors are computed through an iterative process via simple multiplicative updates which do not require learning rate tuning, operate in parallel, and converge monotonically to a global minimizer of the $l_{p,q}$-norm.

The second contribution is an analysis of the optimization landscape for learning in these networks in the vein of gradient flows. The study outlines a weight-balancing flow where the regularizer remains minimal with respect to rescaling transformations as the weights descend in the loss function. This process deviates from ordinary gradient flow for $l_2$-norm regularization, suggesting a more nuanced connection between alternative flows and regularizers. These results build on the existing understanding of deep learning networks, highlighting the significant role of rescaling symmetries and their influence on weight balancing and learning optimization, and providing insights into analogous gauge-fixing conditions for neural networks similar to those found in physical sciences.

**Audience:**

Yes

**Claims And Evidence:**

No

**Requested Changes:**

The main requested changes that I have are the following:

- Please provide empirical evidence that the weights remain balanced when they are initialized in a balanced state.
- Please provide empirical evidence of exponential convergence when weights are not initialized in a balanced state.
- Please increase the quality of the proof in Prop 2.2
- Please consider shortening the paper to 12 pages and including main proofs in the appendix and providing proof sketches if necessary.
- Please provide empirical evidence of the speed of learning after initializing weights with Algorithm 1 in a head-to-head comparison with Stock et. al 2019.
- Please update the theorem statement in 3.4 to accurately include the proper assumptions on the Hessian.

**Strengths And Weaknesses:**

**Strengths**
This paper provides some interesting ideas about the use of gauge transformations in feedforward neural networks. In particular, the examination of rescaling symmetry and how to use them to invoke a minimum norm solution to the weights of a network are an interesting direction of research. Furthermore, the insight that weight-balancing gradient flows if initialized properly remain balanced throughout training is useful insight and motivates the development of regularizers that effectively use the rescaling symmetry.

**Presentation Weaknesses**

This paper has 21 pages of main content which is significantly larger than the 12-page submission of a typical TMLR paper. This would be acceptable if there were 21 pages worth of content. This in my humble opinion is not the case. The paper is too verbose and expository and the amount of information does not justify its length. Specifically, all proofs could easily be placed in the appendix and instead could be replaced with proof sketches if necessary. Secondly, there is really no need to provide proof for corollaries---certainly not in the main text---as these should be straightforward applications of the main theorems.

I also find the organization of the paper unnecessarily confusing in section 2. I'm not sure why the proof of Theorem 2.1 does not immediately follow. The authors claim to build intuition but they also provide analysis of the fixed points and it takes a much longer time to reach the proof. Either put the proof in the appendix if its not important and provide a proof sketch or provide a more linear path in section 2. This arbitrary suspense is not necessary.

**Technical Weaknesses**
There are several technical shortcomings in this paper. I will list them in no specific order but all of them are equally important unless specified otherwise.

- There are two claims about the weight balancing flows that need more empirical evidence. The first is that if the weights are initialized in a balanced state they remain balanced throughout. The other main claim is that the regularize converges exponentially fast in cases where the weights are not initialized in a balanced state. Please provide empirical evidence for both these claims on the same type of networks used in Fig 2.

- A central assumption ins Theorem 3.4 (which is not stated in the theorem statement) is that the Hessian is positive definite and thus invertible (i.e. eqn 48). In general, this is not true. Nothing suggests that the Hessian matrix will be positive definite and thus I find it misleading to follow the conclusions of this theory as this is not a general statement at all. For instance, one could invoke strong convexity to guarantee the Hessian is positive definite but I do not think you do this. So at present the theorem statement and the assumptions in the proof do not match and that needs to be updated. This is also true for Corollary 3.5

- One of the limited motivations of this work is the speed of learning achieved after rescaling transformations. For instance, Stock et. al 2019 motivate their work through minimum norm solutions that learn faster and also generalize better and they show this empirically. This work borrows the same motivations but does not follow through with empirical evidence. Specifically, these claims are not empirically corroborated for the particular type of Multiplicative weight update presented in the paper.

- The paper also leans heavily on symmetry and conservation laws. This is used to motivate the gradient flow through a conservation law perspective but this is not explored at all. What is the point of bringing up conservation laws if you wont to use them for something?

- Prop 2.2 proof is not complete proof. The inductive hypothesis and induction step could be written more clearly. Furthermore, I think this proof of the proposition could be generalized. It appears that all you need is the ratio $a_i/a_j$ to be constant $\forall i,j$ not that $a_i=a_j=1$. I think the reason this proof goes through is by construction that $a_j$ for all non-hidden units is set to 1. But I'm not 100% convinced that there cannot be a fixed point that is not $a_i=1$ but instead has the same ratio constant. Please correct me if I'm wrong here by providing a mathematical argument.

- Updates in eqn 8-10 can be parallelized but it seems that this must happen one layer at a time in a layer-wise fashion. Otherwise, a hidden layer below and its rescaling factor can affect the current layer's rescaling factor. This means that the overall algorithm is bottlenecked by the depth of the network, not width, and is thus not fully parallelizable. Is this the correct interpretation?



**Minor**
- Typo: fA's (right above Proof by Lemma 3.2 on Page 16)

---

> ### Author Response · Authors · 2023-07-24
> **Reply to Reviewer LFZ9**
>
> Thanks very much for your time and consideration.
>
> * You state that "a central assumption in Theorem 3.4 (which is not stated in the theorem statement) is that the Hessian is positive definite ..."
>
> We are baffled by the insinuation that we are attempting to mislead the reader on this point. Here is the verbatim text of the theorem: "Then the weight balancing flow in eq (40) descends everywhere that the loss function is not stationary with respect to $\omega$ and  the regularizer has a positive definite Hessian with respect to $\omega$." This qualification also appears explicitly in eq. (45).
>
> * You state that "In general this is not true. Nothing suggests that the Hessian matrix will be positive definite ..."
>
> Note that we are discussing the Hessian of the network's ${\bf regularizer}$, not its loss function; see the footnote on page 17. In addition, corollary 3.8  establishes this property for the entire family of regularizers based on $\ell_{p,q}$-norms. (See page 19: "The proof requires two steps---first, to compute the Hessian, and second, to show that it is positive-definite.")
>
> * You state that Proposition 2.2 does not provide a complete proof and relies by construction that $a_j=1$ for all non-hidden units.
>
> There are no rescaling degrees of freedom at non-hidden units, so it is appropriate for the Algorithm to fix $a_j=1$ at all input and output units. The lemma is only needed to characterize the fixed points that can be discovered by the Algorithm.
>
> Even if we were to allow $a_i=5$ (say) for all units, any non-zero biases of the network (transforming as $b_i\leftarrow b_i a_i$) would not be invariant under such a transformation.
>
> Can you explain which step of the induction is unclear? The goal is to prove that $a_i=1$ for all units. Here is the proof simplified for a network with three units and two weights ($W_{21}$ and $W_{32}$). The base case is that $a_1=1$ because unit one is necessarily an input unit. The weight $W_{21}$ is rescaled by $a_2/a_1$; since $a_1=1$, this weight is only invariant under rescaling if also $a_2=1$. And so on.
>
> * You suggest that the updates are bottlenecked by the depth of the network.
>
> This understanding is incorrect. The algorithm minimizes an auxiliary function in which the rescaling factors at different units are completely decoupled. This decoupling is achieved by the inequality of geometric and arithmetic means in eqs (22-23) and (27-28); note how the ratio $(a_i/a_j)^r$ in these equations is replaced by $a_i^{2r} + a_j^{-2r}$.
>
> * You ask for empirical evidence that the weights remain balanced if they are properly initialized and that otherwise they converge exponentially to a balanced state.
>
> These statements are rigorously proven under the idealization of certain flows. The balance at each unit is characterized by the condition $Q_i=0$, as in eq (41), and it is  shown that $dQ_i/dt=-\lambda Q_i$. Have you identified a flaw in this proof, or can you otherwise elaborate on the need for empirical evidence?
>
> * You say that "the paper leans heavily on symmetry and conservation laws ... What is the point of conservation laws if you won't use them for something?"
>
> A weight-balancing flow is precisely one in which there is a conserved quantity equal to zero (measuring the balance of weights) at each hidden unit. These flows are derived by appealing to the network's rescaling symmetries. These ideas could not be more tightly linked.
>
> * You object to the section that builds intuition for the fixed points of Algorithm 1. But these results are also used later to prove convergence; they are not merely decorative.
>
> * You state that the paper could be shortened by placing all the proofs in the appendix, only providing proof sketches in the main text, and removing the proofs of corollaries altogether. You suggest this to allow for further empirical investigation of weight-balancing flows.
>
> The paper presents a theoretical study of deep learning, and its main contributions are the careful statements and rigorous proofs of its theorems. The paper was submitted to TMLR so that these proofs could be verified by peer review, not relegated to un-reviewed appendices. The lemmas and corollaries are not necessarily trivial; they are only labeled as such so as to highlight the main theorems. We are also interested in further empirical investigations, but these will require different authors with different expertise. A theoretical study seems within the purview of TMLR as entire conferences (e.g., COLT, DeepMath) are devoted to such work. For such papers, TMLR asks reviewers to assess the paper's technical correctness, to establish that the results are not derivative of other work, and to judge whether the results may be of interest to some in the field (even if  practitioners might view them as incremental or incomplete). Finally, TMLR explicitly allows longer submissions in exchange for longer reviewing periods.
>
> Thanks for your consideration of these points.

---

> > ### Comment · Reviewer_LFZ9 · 2023-08-22
> > **Re:Rebuttal**
> >
> > I thank the authors for their time in writing the rebuttal.
> >
> > I retract my previous criticism of theorem 3.4. I missed the footnote and now the statement makes sense. However, it would be good to have this stated more directly in the paper rather than in a footnote.
> >
> >
> > The rest of the paper's criticisms are minor. My main criticism, which, unfortunately, still remains is the lack of experiments. This would be fine if there was not a precedent in the literature but all close related work have experiments to back their theory and as a result a similar standard must be applied here---despite how interesting the theory might seem.

---

> > > ### Author Response · Authors · 2023-08-22
> > > **Contributions for TMLR**
> > >
> > > Thanks again for your time and consideration. We appreciate the extra commitment required to review a paper of this length.
> > >
> > > As you suggest, we can state the nature of the Hessian directly in the paper in addition to the footnote.
> > >
> > > We believe that you are asking for experimental evidence that weight-balancing leads to (say) faster learning or better generalization. It is true that such evidence has been provided for the case $p=q=2$; it is also true that this case reduces to gradient descent, so that it is much more straightforward to investigate by current infrastructures for deep learning.
> > >
> > > We agree with your suggestion that such evidence would lend greater significance to the theorems in the paper. We disagree with the suggestion that such evidence is needed to validate the technical correctness of these theorems. The theorems claim that the weights in neural networks can be initially balanced by multiplicative updates and indefinitely balanced by certain idealized flows. These claims are supported by mathematical proof, and the steps in these proofs have not been disputed.
> > >
> > > You suggest that the standards of TMLR, as imposed on previous work, require these supplementary experiments despite "how interesting the theory might seem."
> > >
> > > We have attempted to cite all relevant papers on this subject, but in fact, none of the cited papers appear in TMLR. TMLR is a relatively new venue with its own mission statement and acceptance criteria. From the TMLR web page:
> > >
> > >  * "TMLR emphasizes technical correctness over subjective significance, to ensure that we facilitate scientific discourse on topics that may not yet be accepted in mainstream venues but may be important in the future."
> > >
> > >  * "The most important criteria ... assessing the technical soundness as well as the clarity of the narrative."
> > >
> > >  * "We explicitly avoid these terms (“significant”, “impactful”, “novel”), and focus instead on the notion of “interest. If the authors make it clear that there is something to be learned by some researchers in their area from their work, then the criteria of interest is considered satisfied."
> > >
> > > Thanks again for your comments and consideration.

---

### Decision · Action_Editors · 2023-08-24

**Recommendation:** Accept with minor revision

**Comment:**

Following the rebuttal, reviewers generally agreed that the contribution of this manuscript, theoretical in nature, is sound and of interest to at least part of the TMLR community.  The main criticisms brought forth are as follows:

(1) lacking empirical evaluation and practical utility of the established theory; and

(2) overly long presentation which may burden readers unnecessarily.

The authors argue that per the TMLR guidelines, point (1) should not be an impediment for publication $-$ a statement which I (as well as most reviewers) accept.  With regards to point (2), I would like to ask the authors to follow a practice which in my opinion has become the standard in ML $-$ differentiation between primary content placed in paper body, and secondary content located in appendixes.  The latter for example may include complete proofs, with short proof sketches appearing instead in paper body.  Following a skim through the text, I agree with some of the reviewers that there is no reason for the main content to exceed the 12 page mark of TMLR.

Overall, given that point (1) is dismissed and point (2) can be treated easily, I am recommending to accept the paper subject to a minor revision.

**Audience:**

There are members of the TMLR readership for whom the contribution may be of interest

**Claims And Evidence:**

Claims in the paper are well supported by evidence